# Amplification of USP13 drives ovarian cancer metabolism

Cecil Han[1,*], Lifeng Yang[2,*], Hyun Ho Choi[1,*], Joelle Baddour[2], Abhinav Achreja[2], Yunhua Liu[1], Yujing Li[1], Jiada Li[3], Guohui Wan[1], Cheng Huang[4], Guang Ji[5], Xinna Zhang[6], Deepak Nagrath[2,7] & Xiongbin Lu[1]

Dysregulated energetic metabolism has been recently identified as a hallmark of cancer. Although mutations in metabolic enzymes hardwire metabolism to tumourigenesis, they are relatively infrequent in ovarian cancer. More often, cancer metabolism is re-engineered by altered abundance and activity of the metabolic enzymes. Here we identify ubiquitin-specific peptidase 13 (USP13) as a master regulator that drives ovarian cancer metabolism. USP13 specifically deubiquitinates and thus upregulates ATP citrate lyase and oxoglutarate dehydrogenase, two key enzymes that determine mitochondrial respiration, glutaminolysis and fatty acid synthesis. The *USP13* gene is co-amplified with *PIK3CA* in 29.3% of high-grade serous ovarian cancers and its overexpression is significantly associated with poor clinical outcome. Inhibiting USP13 remarkably suppresses ovarian tumour progression and sensitizes tumour cells to the treatment of PI3K/AKT inhibitor. Our results reveal an important metabolism-centric role of USP13, which may lead to potential therapeutics targeting USP13 in ovarian cancers.

[1] Department of Cancer Biology, The University of Texas MD Anderson Cancer Center, Houston, Texas 77030, USA. [2] Department of Chemical and Biomolecular Engineering, Rice University, Houston, Texas 77005, USA. [3] The State Key Laboratory of Medical Genetics and School of Life Sciences, Central South University, Changsha 410008, China. [4] Drug Discovery Laboratory, School of Pharmacy, Shanghai University of Traditional Chinese Medicine, Shanghai 201203, China. [5] Institute of Digestive Diseases, Longhua Hospital, Shanghai University of Traditional Chinese Medicine, Shanghai 200032, China. [6] Department of Gynaecologic Oncology and Reproductive Medicine, The University of Texas MD Anderson Cancer Center, Houston, Texas 77030, USA. [7] Department of Bioengineering, Rice University, Houston, Texas 77005, USA. * These authors contributed equally to this work. Correspondence and requests for materials should be addressed to D.N. (email: deepak.nagrath@rice.edu) or to X.L. (email: xlu2@mdanderson.org).

Cancer cell proliferation requires abundant building blocks and energy to fulfil their cell growth and division[1–3]. To meet these elevated requirements, cancer cells undergo major modifications in their metabolic pathways[4–7]. Steady supply of metabolic intermediates generated from tricarboxylic acid (TCA) cycle is required to synthesize macromolecules such as lipids, nonessential amino acids and nucleotides[1,4,8–10]. Therefore, anaplerosis is crucial to keep the availability of anabolic precursors and replenish the TCA cycle[1,4,8–10]. Glutamine, as the major anaplerotic precursor for mitochondrial oxaloacetate, is required to boost the maximal mitochondrial metabolism in rapidly dividing cells[4,6,8,10–14]. In a recent study, we reported that high-invasive ovarian cancer (OVCA) cells are markedly glutamine dependent[4]. Mitochondrial utilization of glutamine begins with a two-step conversion of glutamine to α-ketoglutarate (α-KG or 2-oxoglutarate), typically by glutaminase and glutamate dehydrogenase[15,16]. α-KG can be either oxidized by oxoglutarate dehydrogenase (OGDH) to succinate or reductively carboxylated by isocitrate dehydrogenase to isocitrate and then citrate[15,17,18]. Glutamine-derived citrate is transported to the cytoplasm to generate acetyl-CoA for fatty acid synthesis[17,19].

In rapidly proliferating cancer cells, citrate is generated by the TCA cycle either from glucose by glycolysis or from glutamine by anaplerosis[20]. ATP citrate lyase (ACLY) is most abundantly expressed in the liver and white adipose tissue, while it exhibits low expression levels in other tissues[21]. However, ACLY is often upregulated or activated in human cancers, including lung, prostate, bladder, breast, liver, stomach and colon tumours[22]. ACLY inhibitors have been evaluated for their ability to block fatty acid synthesis and cancer cell proliferation, among which SB-204990 was shown to be effective in both *in vivo* and *in vitro* tumour models[23].

To replenish TCA cycle intermediates and sustain anabolic processes, cancer cells rely excessively on glutamine, which enters the TCA cycle as α-KG via the α-ketoglutarate dehydrogenase (KGDH) complex[6,8,13]. In cancer cells, glutamine uptake is markedly enhanced and far exceeds the metabolic requirements of the cell. α-KGDH is a complex enzyme consisting of three types of subunits, including OGDH (OGDH, E1), dihydrolipoamide succinyltransferase (DLST, E2) and dihydrolipoamide dehydrogenase (DLD, E3)[24,25]. Through OGDH and other TCA cycle enzymes, α-KG, generated from glutamine typically by glutaminase and glutamate dehydrogenase, can also be oxidized into oxaloacetate with plenty of NADH generation[26]. Except for the cellular duplication requirement, it has been recently reported that mitochondrial glutamine oxidation is essential for OVCA cell metastasis[27]. As the E1 subunit of α-KGDH complex, OGDH is a rate-limiting component for the overall conversion of α-KG to succinyl-CoA and $CO_2$ (ref. 28). Increase levels of OGDH are able to rewire cell metabolism and promote tumourigenesis. As examples, upregulation of OGDH was identified as a driver for hepatocellular carcinoma[29]. Recent studies also reported that targeting the α-KGDH complex inhibits amino-acid metabolism and regulates oxidative stress in cancer cells[13,30]. Inhibition of OGDH leads to buildup of lactic acid and suppresses cell growth[15,31]. Oxidation of α-KG is required for reductive carboxylation in cancer cells with mitochondrial defects[31], which is likely suppressed by OGDH inhibition.

High-grade serous ovarian cancer (HGSC) is the most lethal cause of gynaecological cancer deaths[32]. Not much improvement has been achieved in overall survival for patients with HGSC in the past 30 years, and standard therapy has not advanced beyond platinum-based combination chemotherapy[33]. Genomic analyses reveal that HGSC is driven primarily by gene copy-number changes rather than recurrent gene point mutations, among which genomic alteration of the PI3K/AKT pathway is most common and associated with poor clinical outcomes[33–35]. Here we determined molecular interactions between genes in the ubiquitin–proteasome pathways and cancer cell metabolism. We identified genomic amplification of USP13 in HGSC and the association of USP13 overexpression with aggressive OVCA progression. USP13 upregulates ACLY and OGDH, two key regulators that drive glutaminolysis, ATP generation and lipid synthesis in cancer metabolism. Inhibition of USP13 simultaneously suppresses glutamate anaplerosis to refill the TCA cycle and the generation of acetyl-CoA, a vital building block for *de novo* biosynthesis of fatty acids, leading to the marked supersession of OVCA cell proliferation and tumourigenic potential. These findings may lead to the development of USP13 inhibitors and new-targeted therapies in OVCAs.

## Results

**The *USP13* gene is frequently amplified in human OVCA.** Deubiquitinases play a central role in regulating protein stability and activity[36]. Our analysis of cancer genomics revealed that gene mutation is infrequent in deubiquitinase genes. However, *USP13* gene is amplified in several types of human cancer, particularly in 52% of lung squamous cell carcinoma and 29.3% of HGSC (Fig. 1a; Supplementary Fig. 1a). Genomic analyses of 538 human HGSC and 580 lung squamous cell carcinoma in The Cancer Genomics Atlas (TCGA) and high-resolution single-nucleotide polymorphism-based copy-number databases (Affymatrix SNP6 and Nexus Copy Number program; Supplementary Fig. 1b) demonstrated the *USP13* gene-containing amplicon across chromosome 3q26.2–3q26.3. Interestingly, the *USP13* gene is adjacent to *PIK3CA* (phosphatidylinositol-3-kinase catalytic subunit, α-isoform) in the amplicon (Fig. 1b; Supplementary Fig. 1b). A perfect positive correlation exists between copy numbers of *USP13* and *PI3KCA* ($R = 0.9722$, $P < 0.0001$, unpaired *t*-test), confirming their co-amplification in OVCA (Fig. 1c). Copy number of *USP13* is also significantly correlated with its messenger RNA expression ($R = 0.306$, $P = 0.0002$, unpaired *t*-test) (Fig. 1d). In the immunohistochemical (IHC) analysis using a tumour tissue microarray including 118 OVCA samples and 16 adjacent normal tissues (Fig. 1e), expression of USP13 was scored according to the staining intensity and proportion of signals in each sample (Supplementary Fig. 1c). In comparison with normal ovarian tissues, OVCA samples have ~3.7-fold higher expression levels of USP13 ($P = 0.0006$; Fig. 1e). Aberrant overexpression of USP13 in ovarian tumours was also confirmed by western blot using normal and tumour tissue samples (Fig. 1f; Supplementary Fig. 2). USP13 expression is markedly increased in advanced ovarian tumours and is significantly correlated with tumour grade (Fig. 1g). Consequently, high expression of USP13 is significantly associated with poor survival of patients with OVCA (Fig. 1h).

**Inhibiting USP13 suppresses OVCA tumourigenic potential.** Frequent amplification of *USP13* was also identified in 53 human OVCA cell lines from the Cancer Cell Line Encyclopedia (Fig. 2a). Copy numbers and protein expression of *USP13* were validated by quantitative PCR and immunoblotting in seven lines (Fig. 2b; Supplementary Fig. 3a). CAOV3, HeyA8 and OAW28 are *USP13*-amplified cell lines, while IGROV1 and SKOV3 have no *USP13* amplification. *USP13* is not amplified but highly expressed in OVCAR8, implicating regulatory mechanisms for USP13 expression other than gene amplification in the OVCAR8. Stable knockdown (KD) of USP13 by lentiviral short hairpin RNAs (shRNAs) markedly inhibited the proliferation of the

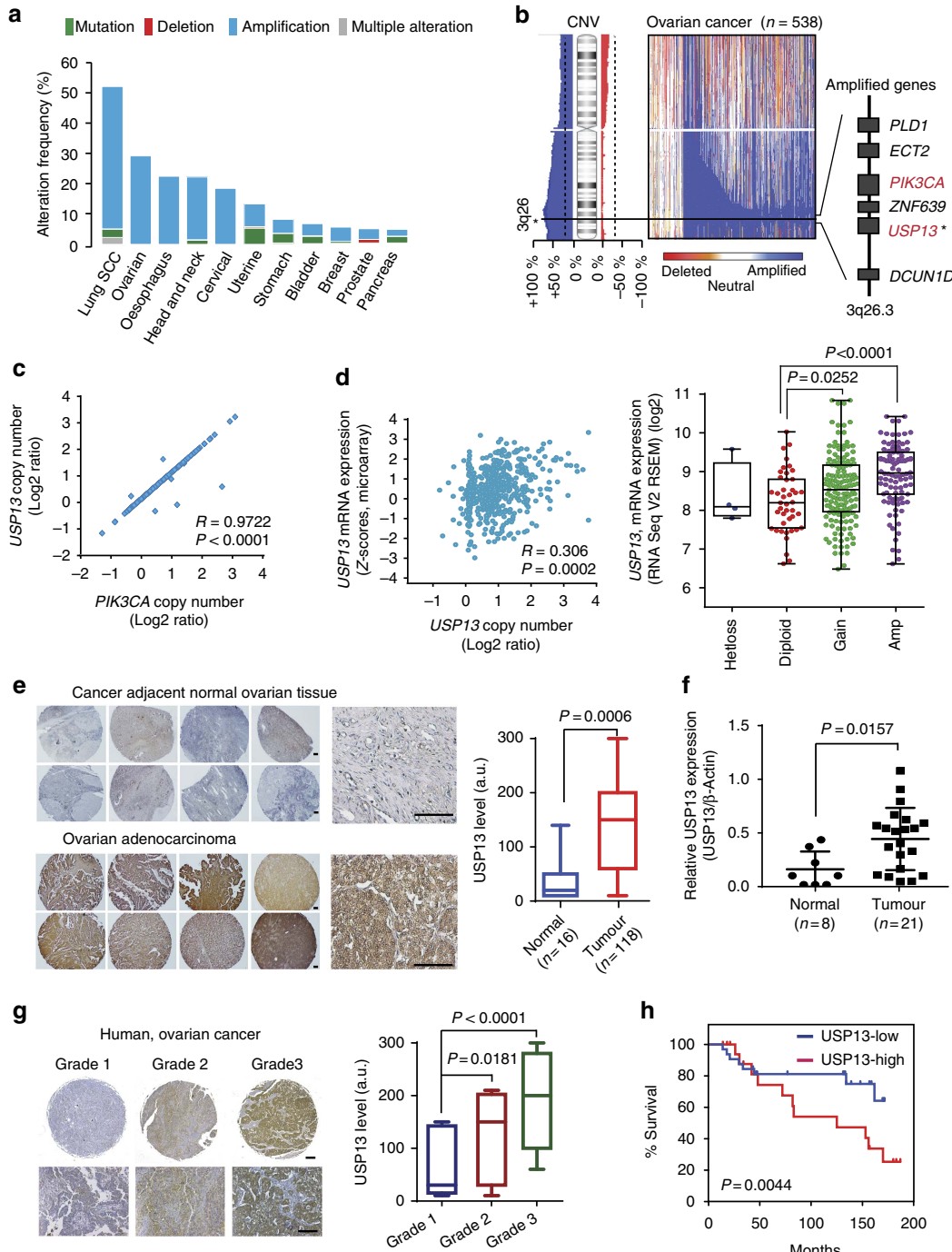

**Figure 1 | Genomic amplification of *USP13* is positively correlated with ovarian cancer progression.** (**a**) Genomic alterations of *USP13* across human cancers determined by cBioportal analysis of TCGA databases. (**b**) Integrated analysis of USP13 amplification in 538 HGSC samples. Frequency plots of the copy-number abnormalities indicate degree of copy-number loss (red) or gain (blue). The colour intensity indicates the extent of copy-number changes. Representative genes in the amplicon of chromosome 3q26.2–3 are shown. (**c**) Positive correlation between *USP13* and *PI3KCA* copy numbers. (**d**) Scatterplots of USP13 copy number versus messenger RNA expression in OVCA (left panel). USP13 expression levels are positively correlated with its gene copy numbers (right panel). (**e**) USP13 is highly expressed in ovarian tumours. Representative IHC-staining images of USP13 in a tissue microarray of OVCA and adjacent normal ovarian tissues (scale bar, 100 μm). Relative USP13 expression levels in ovarian tumours ($136.7 \pm 11.29$, $n = 118$) were compared with that of normal tissues ($37.27 \pm 13.15$, $n = 16$). (**f**) Western blot analysis of USP13 expression in normal ovarian tissues ($n = 8$) and ovarian cancer tissue samples ($n = 21$). USP13 blots were quantified by phosphoimaging and normalized to the levels of β-actin. (**g**) USP13 expression levels are positively correlated with ovarian tumour progression. Grade 1 ($n = 18$), grade 2 ($n = 24$) and grade 3 ($n = 38$). Representative IHC-staining images of tissue samples are shown. (**h**) High expression of USP13 is associated with poor overall survival of patients with OVCA. Kaplan–Meyer plots for overall survival are shown. Tumour samples were divided into USP13 low group ($n = 84$, intensity score from 10 to 150) and USP13 high group ($n = 84$, score from 150 to 300). Box and whiskers indicate minimum to maximum percentiles in this figure: centre line, median value; upper box limit, 75% percentile; lower box limit, 25% percentile; whiskers, minimum or maximum values. Signals of immunohisto-chemistry data in tumour cells were visually quantified using a scoring system from 1 to 3, multiplied intensity of signal and percentage of positive cells (signal: 0 = no signal, 1 = weak signal, 2 = intermediate signal and 3 = strong signal; percentage: 10–100%). a.u., arbitrary unit. Unpaired *t*-test was used for statistical analyses in **c**, **d**, **e**, **f** and **g**. Log-rank (Mantel–Cox) test was used to analyse *P* value in **h**.

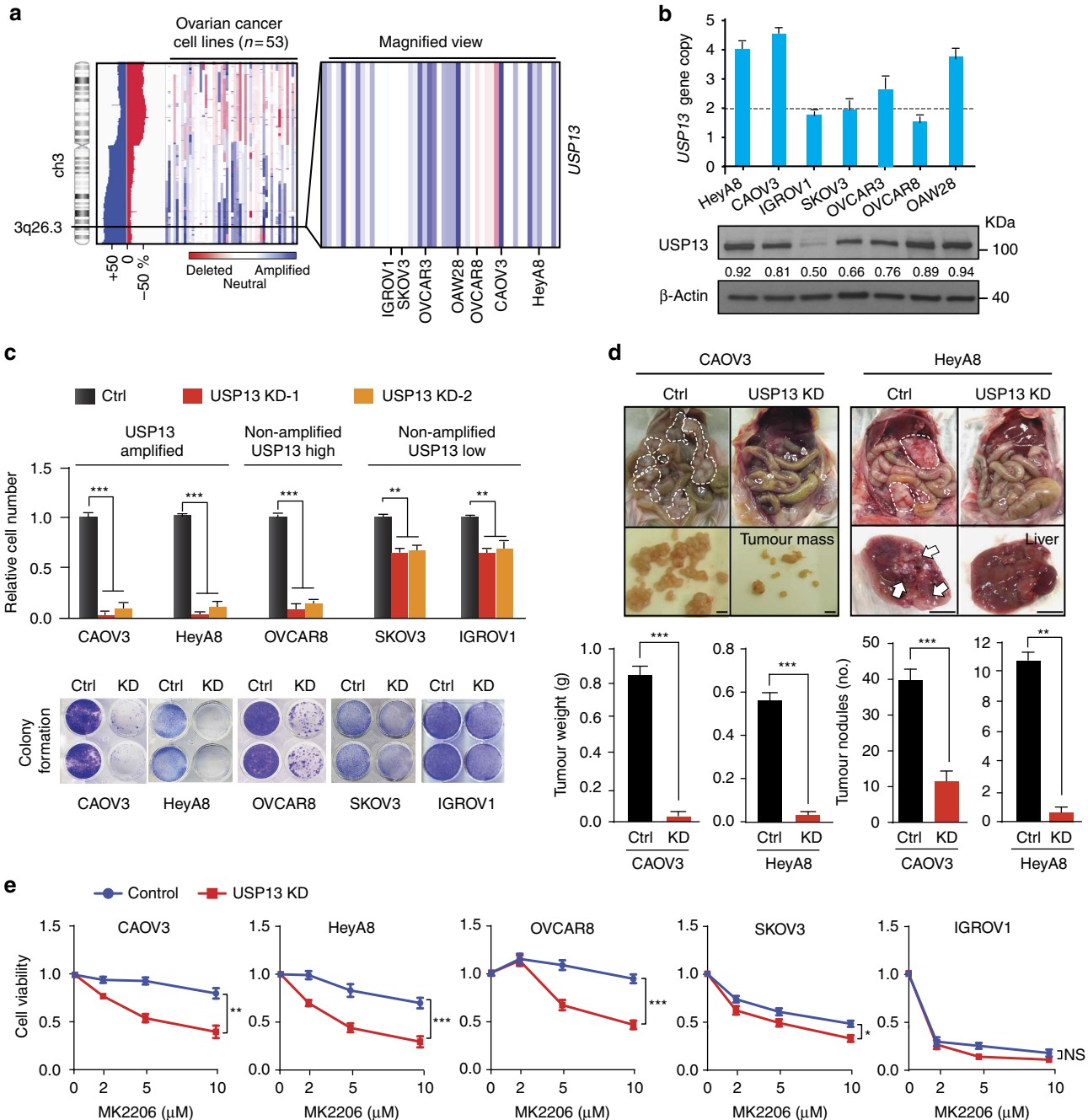

**Figure 2 | Inhibiting USP13 suppresses OVCA cell proliferation and tumourigenic potential.** (**a**) Genomic alteration frequency analysis shows USP13 amplification in 53 human OVCA cell lines. The colour intensity and copy-number changes are shown as described in Fig. 1b. (**b**) Copy number and protein expression of USP13 in seven representative cell lines were validated by quantitative PCR and immunoblotting. USP13 blots were quantified by phosphoimaging and normalized to the levels of β-actin. (**c**) Relative proliferation of OVCA cells stably expressing control or USP13 shRNAs. Control scramble shRNA was used as a negative control. Cells were cultured for 7 days with expression of shControl or shUSP13 and then stained by crystal violet. Representative crystal violet staining of OVCA cells were shown at the bottom. Unpaired *t*-test was used for statistical analysis. ($n = 3$). (**d**) Tumour formation in mice transplanted with CAOV3 or HeyA8 cells expressing control or USP13 shRNAs. Mice were assessed at week 7 for CAOV3 and week 5 for HeyA8 ($n = 6$ per group). Representative images show tumour growth and pattern of spread. Scale bars, 0.5 cm. Tumour weights and numbers of tumour nodules are shown at the bottom. Unpaired *t*-test was used for statistical analyses. (**e**) Depletion of USP13 sensitizes OVCA cells with high USP13 expression to the treatment of the AKT inhibitor, MK-2206. Unpaired *t*-test was used for statistical analysis.*$P < 0.05$, **$P < 0.01$, ***$P < 0.001$, NS, nonsignificant. Error bars represent ± s.d. in this figure.

*USP13*-amplified cell lines (CAOV3 and HeyA8) and the USP13-overexpressing cell line (OVCAR8). By contrast, KD of USP13 only modestly reduced proliferation of cell lines with low USP13 expression (SKOV3 and IGROV1) (Fig. 2c; Supplementary Fig. 3b,c). In two other USP13 high HGSC cell lines (OAW28 and OVCAR3)[37], silencing USP13 also profoundly inhibited cell proliferation (Supplementary Fig. 3d). The results suggest that OVCA cells are probably addicted to the USP13 amplification or overexpression. Next, we determined the effect of USP13 on the tumourigenic potential of OVCA cells. Depletion

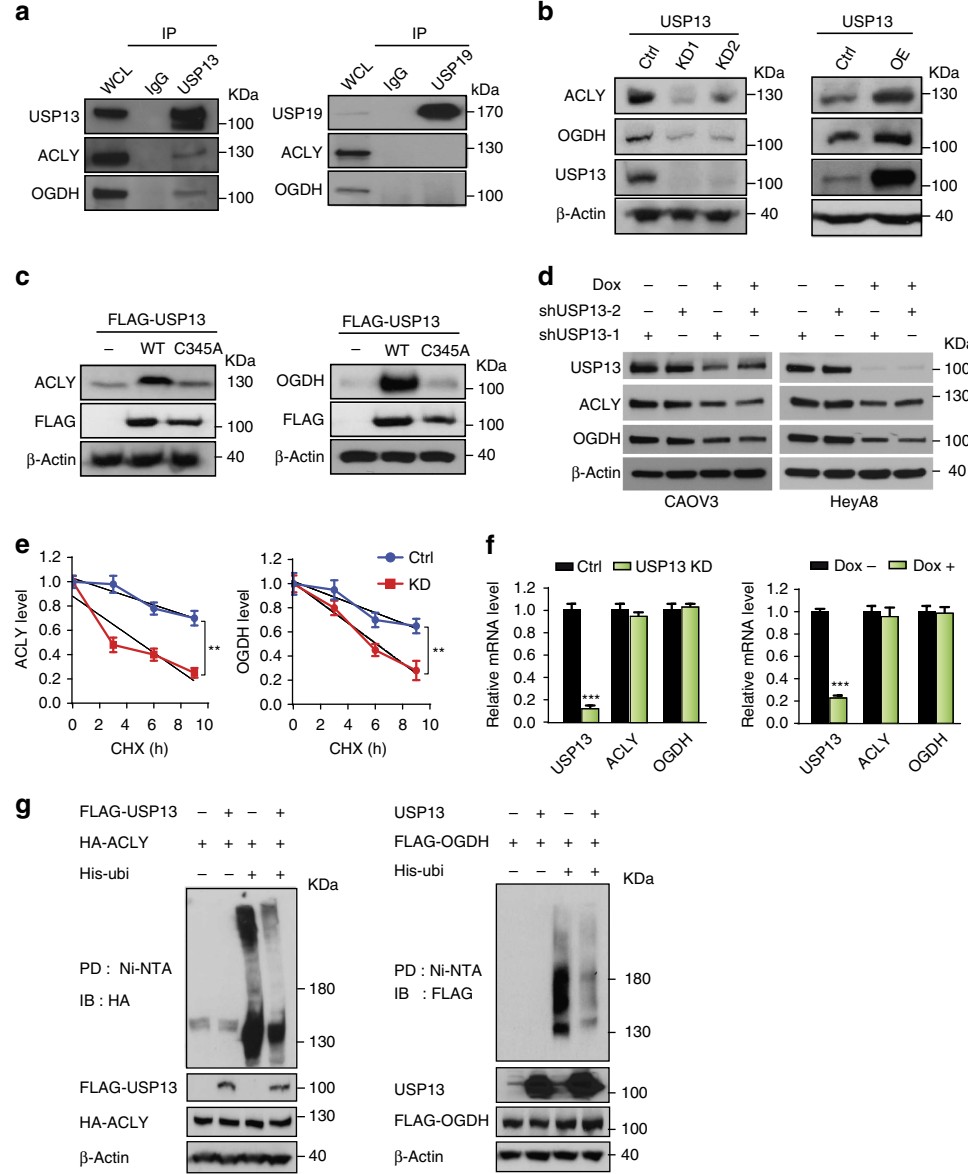

**Figure 3 | USP13 stabilizes ACLY and OGDH.** (**a**) USP13 physically interacts with ACLY and OGDH. Immunoprecipitation (IP) and western blot analyses were performed using indicated antibodies. Normal IgG was used as a negative control for IP. USP19 was used as a negative control DUB. WCL, whole-cell lysate. (**b**) Knockdown (KD) of USP13 decreases endogenous levels of ACLY and OGDH (left panel), while overexpression (OE) of USP13 increases their levels (right panel). (**c**) Enzymatically dead mutant of USP13 (C345A) fails to stabilize ACLY and OGDH. (**d**) Dox-induced USP13-KD decreases the levels of ACLY and OGDH in CAOV3 and HeyA8 cells. (**e**) Protein stability of ACLY and OGDH is reduced by USP13-KD. CAOV3 cells expressing control or USP13 shRNA were treated with cycloheximide (CHX, $100\,\mu g\,ml^{-1}$) for indicated time points ($n=3$). Relative levels of ACLY and OGDH were quantified as a percentage of initial protein level. (**f**) USP13-KD does not affect the transcription of ACLY and OGDH ($n=3$). Unpaired $t$-test was used for statistical analysis. (**g**) USP13 overexpression induces deubiquitination of ACLY or OGDH. HEK293T cells were co-transfected with indicated expression vectors and treated with $5\,\mu g\,ml^{-1}$ MG132 for 6 h before they were collected. His-tagged ubiquitin (His-ubi) were pulled down (PD) through nickel-charged magnetic agarose beads (Ni-NTA). Ubiquitinated proteins were PD and analysed by immunoblotting (IB) with anti-HA or anti-FLAG antibodies. Error bars represent ± s.d. in this figure. $^{**}P<0.01$, $^{***}P<0.001$.

of USP13 resulted in profound decrease in tumour weight and number of tumour nodules in both CAOV3- and HeyA8-derived ovarian tumour xenografts (Fig. 2d).

The PI3K/AKT/mTOR pathway is the most frequently altered in OVCA[33,35]. Since *USP13* and *PIK3CA* are often co-amplified, we asked whether the *USP13* amplification influences the sensitivity of OVCA cells to the inhibition of PI3K/AKT. We found that KD of USP13 sensitized CAOV3, HeyA8 and OVCAR8 cells to the treatment of a pan-AKT inhibitor, MK-2206 (Fig. 2e; Supplementary Fig. 3e). By contrast, SKOV3 and IGROV1 cells expressing low levels of USP13 and PIK3CA

were very sensitive to the treatment of MK-2206- and USP13-KD had minimal effect on cell survival. The results suggest that inhibition of USP13 may enhance the effects of PI3K/AKT inhibitors on killing OVCA cells.

**ACLY and OGDH are deubiquitination substrates of USP13.** We performed mass spectrometry to analyse the USP13-containing protein complex. ACLY and OGDH were identified as putative USP13-interacting proteins (Supplementary Fig. 4a). Western blot analyses of HEK293T cell lysates immunoprecipitated with USP13 antibody showed the interaction of USP13 with

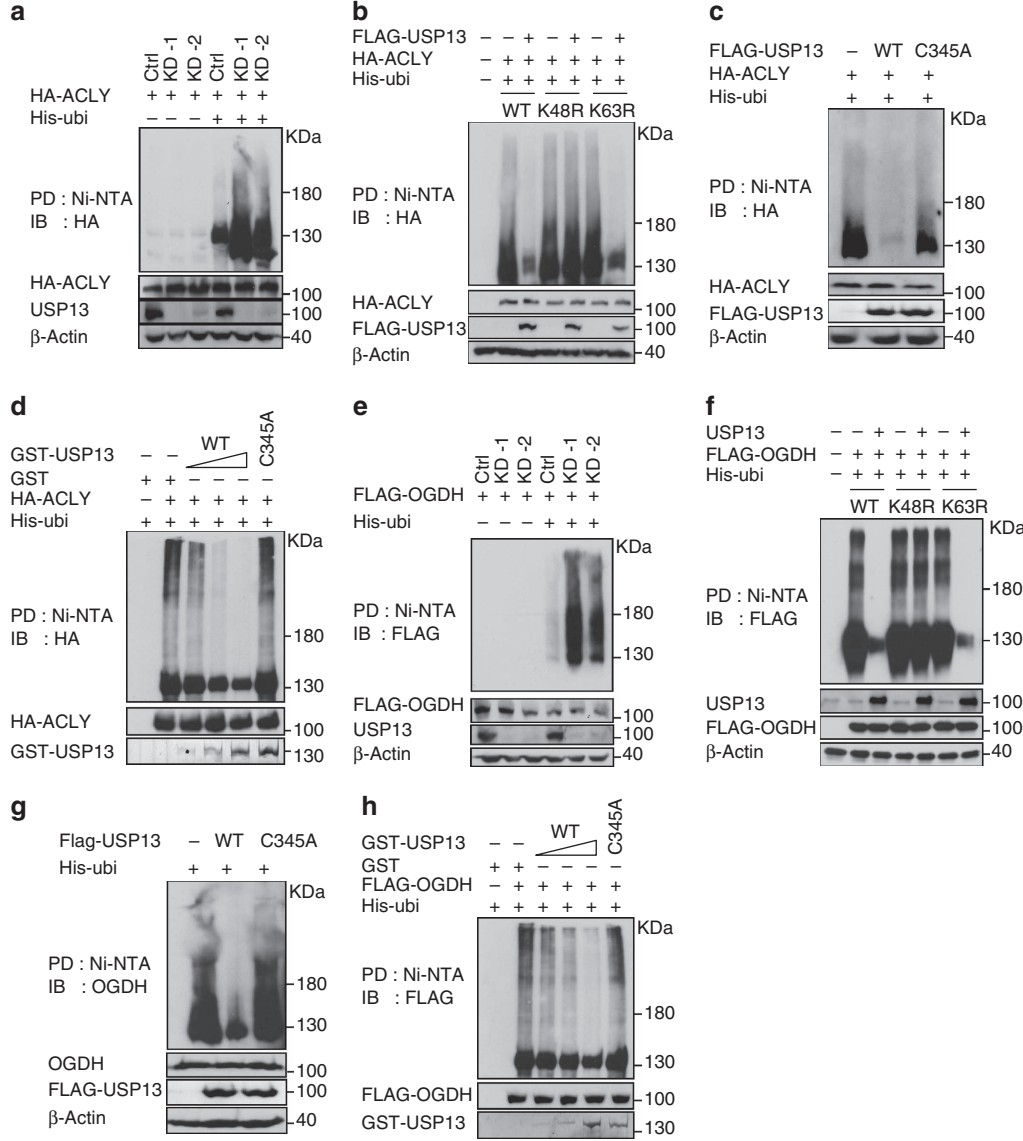

**Figure 4 | USP13 directly deubiquitinates ACLY and OGDH.** (**a**,**e**) USP13 knockdown (KD) increases the level of ubiquitinated ACLY (**a**) and OGDH (**e**). HEK293T cells expressing control and USP13 shRNAs were treated with MG132 and ubiquitinated ACLY or OGDH was pulled down (PD) and subject to immunoblotting (IB) analysis. (**b**,**f**) USP13-mediated deubiquitination acts on the lysine-48 (K48) ubiquitination of ACLY (**b**) and OGDH (**f**). HEK293T cells were co-transfected with the indicated expression vectors. Equal amounts of cell lysates were analysed by immunoprecipitation and IB assays as described above. (**c**,**g**) The C345A mutant form of USP13 fails to deubiquitinate ACLY (**c**) and OGDH (**g**). (**d**,**h**) USP13, but not the C345A mutant, deubiquitinates ACLY (**d**) and OGDH (**h**) in an in vitro deubiquitination assay. HEK293T cells were co-transfected with indicated expression vectors. The cells were treated with 5 µg ml$^{-1}$ MG132 for 6 h before they were collected. Ubiquitinated ACLY or OGDH proteins immunopurified and incubated with varying amounts of bacterially purified GST-tagged USP13 in a deubiquitination buffer. His-tagged ubiquitin (His-ubi) were PD through nickel-charged magnetic agarose beads (Ni-NTA) under the denaturing condition (see Methods) and the ubiquitinated proteins were analysed by IB with anti-HA or anti-FLAG antibodies.

ACLY and OGDH (Fig. 3a). Reciprocal analyses in which immunoprecipitated ACLY or OGDH were probed for USP13 confirmed the USP13–ACLY and USP13–OGDH interactions (Supplementary Fig. 4b). To determine the binding requirements for these interactions, we generated three deletion constructs of USP13 and performed the protein-binding assay (Supplementary Fig. 4c). The N terminus including a ubiquitin-specific protease-type zinc-finger domain (residues 1–336) was essential for USP13 to interact with OGDH, whereas the ubiquitin carboxyl-terminal hydrolase domain (residues 336–863) of USP13 is required for its binding with ACLY (Supplementary Fig. 4c). We postulated that ACLY and OGDH may be the deubiquitinating substrates of USP13. USP13-KD significantly reduced the levels of ACLY and OGDH, while overexpression of USP13 had the opposite effect

(Fig. 3b; Supplementary Fig. 4d). Cys345 of USP13 is a critical residue for its deubiquitination activity, and mutating it to alanine abolishes this activity[38]. We found that the C345A mutant of USP13 was unable to increase the steady-state levels of ACLY and OGDH in contrast to the wild-type USP13 (Fig. 3c).

We generated CAOV3 and HeyA8 cell lines that stably express doxycycline (Dox)-inducible USP13 shRNAs, in which ACLY and OGDH levels were considerably decreased after Dox treatment (Fig. 3d). Next, we examined whether USP13 augments the stability of ACLY and OGDH. Control and USP13-KD CAOV3 cells were treated with cycloheximide (100 µg ml$^{-1}$) to block nascent protein synthesis. Depletion of USP13 markedly decreased protein stability of ACLY and OGDH (Fig. 3e; Supplementary Fig. 4e), but had no effect on the transcription

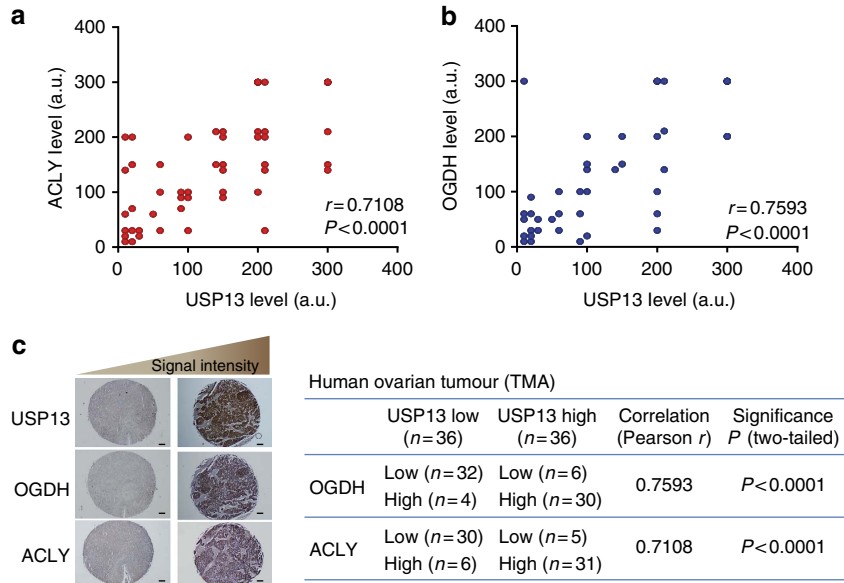

**Figure 5 | USP13 levels are positively correlated with the levels of ACLY and OGDH in ovarian tumours.** (**a,b**) A tissue microarray (TMA) of ovarian tumours were immunostained with antibodies against USP13, OGDH or ACLY, respectively. Immunohistochemical (IHC) signals are scored by multiplying the percentage of positive cells by the staining intensity (Signal: 0 = no signal, 1 = weak signal, 2 = intermediate signal and 3 = strong signal; percentage: 10–100%). a.u., arbitrary unit. Scale bars, 100 μm. Unpaired $t$-test (two-tailed) and Pearson $r$ were used for statistical analysis. (**c**) Positive correlation between USP13 and ACLY or OGDH levels from IHC-staining analysis using human ovary tumour microarray samples.

of ACLY and OGDH (Fig. 3f). Ubiquitination of ACLY and OGDH was assessed in the cells treated with the proteasome inhibitor MG132 that prevents degradation of ubiquitinated proteins. Overexpression of USP13 markedly reduced the levels of ubiquitinated ACLY and OGDH (Fig. 3g), suggesting that USP13 may stabilize ACLY and OGDH through deubiquitination.

The acetyl-CoA derived from citrate also serves as a precursor for protein acetylation[39,40]. Thus, beyond lipid synthesis and glutamine metabolism, USP13 may also influence protein acetylation via upregulation of ACLY. However, KD of USP13 in CAOV3 cells had no notable effects on the levels of global acetylation of histone H3 and H4 (Supplementary Fig. 4f), suggesting that USP13 is not involved in histone protein acetylation. A recent study found that USP13 stabilized PTEN expression by removing PTEN ubiquitination in human breast cancer cells[41]. In OVCA cells, we only observed very modest downregulation of PTEN when USP13 was silenced (Supplementary Fig. 4g). Therefore, USP13 may function in a context-specific manner. It is also noted that USP13 and PIK3CA are not amplified in breast cancer, which may interpret the tumour suppressive functions of USP13 via PTEN in breast cancer.

**USP13 directly deubiquitinates ACLY and OGDH.** Ubiquitin has seven lysine residues as points of ubiquitination, in which K48-linked polyubiquitin chains target proteins for degradation, whereas K63-linked chains are associated with regulatory functions[42]. USP13-KD markedly enhanced the ubiquitination of ACLY and OGDH (Fig. 4a,e). To determine what type of ubiquitination USP13 acts on, we transfected HEK293T cells with K48R or K63R mutant form of ubiquitin. Overexpression of USP13 inhibited wild-type and K63R mutant forms of ubiquitin-associated ubiquitination of ACLY, but had no effect on the K48R ubiquitin-associated ubiquitination (Fig. 4b). The C345A mutant of USP13 failed to reduce the ubiquitination of ACLY (Fig. 4c), suggesting that the deubiquitination activity is essential for the

function of USP13. To exclude the possibility that USP13 indirectly influences the ubiquitination of ACLY in cells, we assessed the *in vitro* biochemical activity of USP13. Increasing amounts of USP13 proteins resulted in decreasing ubiquitination levels of ACLY, while the C345A mutation abolished the activity of USP13 (Fig. 4d). Results from the similar experiments showed that USP13 acts on the K48-associated ubiquitination of OGDH and directly deubiquitinates OGDH (Fig. 4f–h). To further validate the K48-associated deubiquitination activity of USP13, we analysed the ubiquitination of ACLY and OGDH labelled with K48-only or K63-only ubiquitin (all other lysine residues are replaced with arginine residues except for K48 or K63). While neither of ACLY and OGDH were ubiquitinated with K63-only ubiquitin, their K48-assoicated ubiquitination was markedly decreased by USP13 overexpression (Supplementary Fig. 5a,b). These results provide the direct evidence that USP13 possesses specific and intrinsic deubiquitination activity towards ACLY and OGDH. To determine the positive correlation between USP13 and ACLY/OGDH levels in clinical ovarian tumours, we examined their levels using a tumour tissue microarray including 72 clinical ovarian tumour samples. Significantly, positive correlations exist between USP13 and ACLY levels ($r = 0.7108$, $P < 0.0001$; Fig. 5a,c) and between USP13 and OGDH ($r = 0.7593$, $P < 0.0001$) levels (Fig. 5b,c).

**USP13 regulates glutaminolysis and mitochondrial function.** We proposed that KD of USP13 reduces OGDH expression thereby limiting the synthesis of TCA cycle intermediates from oxidative pathways (Fig. 6a). OGDH catalyses α-KG conversion into succinate along with the generation of NADH from $NAD^+$ (refs 24–26). As expected, USP13-KD cells had higher α-KG levels due to the reduced OGDH activity (Fig. 6b). In most cells, glutamine and glucose are the major sources of α-KG. We used [U-$^{13}$C] glutamine (all five carbons are $^{13}$C-labelled, M5) to trace its oxidation into TCA cycle metabolites (Fig. 6a). M5 glutamine is converted into M5 glutamate, which is further converted into

M5 α-KG by glutamate dehydrogenase or transaminases[26]. The M5 α-KG is decarbonized into M4 fumarate, M4 malate and M4 citrate, obtained by condensing of M4 oxaloacetate obtained from glutamine and unlabelled acetyl-CoA from glucose. USP13-KD reduced the percentages of M5 glutamate, M4 fumarate, M4

aspartate, M3 pyruvate, M3 lactate and M3 alanine in HeyA8 and CAOV3 cells (Fig. 6c). Conversely, overexpression of USP13 in the low USP13-expressing SKOV3 cells enhanced glutamine oxidation, as observed through significant increase in percentages of these intermediate metabolites (Supplementary Fig. 6a).

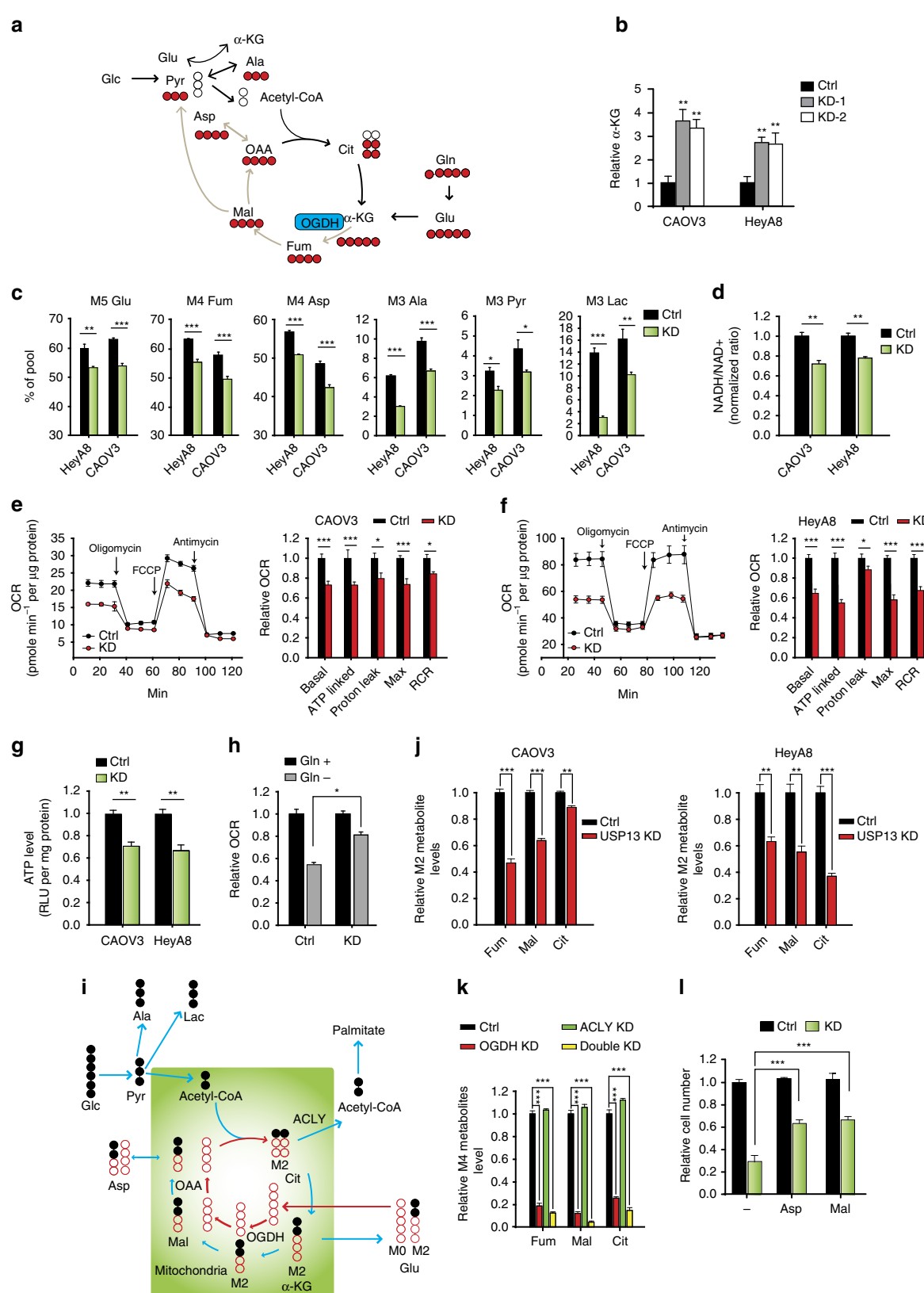

Furthermore, we found that USP13-KD decreased the level of M4 TCA cycle metabolites, which are directly derived from [U-$^{13}C_5$] glutamine, indicating a reduced flux of glutamine into TCA cycle (Supplementary Fig. 6b). Because OGDH plays an important role in NAD$^+$ conversion into NADH, decreased oxidative flux of the TCA cycle led to reduced NADH/NAD$^+$ ratio (Fig. 6d). Marked reduction of basal oxygen consumption rate (OCR) levels was observed in the USP13-KD cells, indicating the influence of USP13 on respiration (Fig. 6e,f). Significant reduction of ATP-linked OCR, maximal OCR and mitochondrial respiratory control ratio were seen in the USP13-KD cells. In support of the involvement of USP13 in bioenergetics, USP13-KD inhibited ATP production in cells (Fig. 6g). Under glutamine depletion conditions, USP13-KD CAOV3 cells are less dependent on glutamine for respiration (basal OCR) compared with control CAOV3 cells (Fig. 6h).

To investigate the effect of USP13 KD on glucose metabolism, we measured glycolytic capacity of OVCA cells when USP13 was knocked down and found that USP13 depletion significantly enhances lactate secretion in CAOV3 and HeyA8 cells, but it does not have a consistent effect on glucose uptake for CAOV3 and HeyA8 (Supplementary Fig. 6c,d). We then used [U-$^{13}C_6$] glucose to trace its contribution to intermediate metabolites within the TCA cycle (Fig. 6i). The [U-$^{13}C_6$] glucose will be converted into M3 pyruvate through glycolysis, and M3 pyruvate will be further converted into M2 acetyl-CoA and M2 TCA cycle metabolites (citrate, α-KG, fumarate and malate). We found that in both CAOV3 and HeyA8 cell lines, USP13-KD decreases the relative levels of M2 fumarate, malate and citrate, which are derived directly from [U-$^{13}C_6$] glucose (Fig. 6j). Therefore, USP13-KD also decreases glucose oxidation into TCA cycle metabolites.

To clarify that the OGDH and ACLY are key metabolic enzymes responsible for the effect of USP13 on TCA cycle metabolism, we knocked down the expression of OGDH, ACLY individually or OGDH and ACLY together. Interestingly, we found that OGDH KD and OGDH/ACLY double KD have profound effects on reducing both glutamine oxidation and glucose oxidation (Fig. 6k; Supplementary Fig. 6e,f). OGDH/ACLY double KD increased glycolysis in both CAOV3 and HeyA8, a similar observation as made in the USP13-KD cells (Fig. 6j; Supplementary Fig. 6e,f). However, ACLY KD alone will not have a significant effect on TCA cycle metabolism in either CAOV3 or HeyA8 cells (Fig. 6k; Supplementary Fig. 6e,f). OGDH is essential for maintaining aspartate, a critical metabolite for cell proliferation[43–45]. Indeed, we observed that supplementing cells with aspartate and malate partially rescued the proliferation of the USP13-KD CAOV3 cells (Fig. 6l). Therefore, we definitively prove that USP13 can regulate mitochondrial TCA cycle metabolism by upregulating OGDH expression level.

**USP13 regulates glutamine-driven reductive carboxylation.** ACLY is a critical enzyme involved in generating acetyl-CoA, a building block for *de novo* lipid synthesis (Fig. 7a)[23]. To determine the role of USP13 in glutamine-dependent reductive carboxylation for lipogenesis, we cultured cancer cells in [U-$^{13}C$] glutamine medium. The reductive carboxylation of M5 glutamine provides M5 α-KG and generates M5 citrate through isocitrate dehydrogenase 1/2 (Fig. 7a)[18]. Citrate is transported out of the mitochondria into the cytoplasm for lipid synthesis. The reductive pathway has been correlated with an increase in the ratio of α-KG to citrate[46]. Indeed, in USP13-KD cells, the ratio of α-KG to citrate was significantly decreased as well as the NADPH/NADP$+$ ratio, indicating reduced reductive flux (Fig. 7b; Supplementary Fig. 7a). In line with our hypothesis, we found that USP13-KD significantly decreased M5 citrate, M3 fumarate and M3 malate pools in CAOV3 cells (Fig. 7c, top panel), whereas overexpression of USP13 had the opposite effect in SKOV3 cells (Fig. 7c, bottom panel).

Since acetyl-CoA derived from glutamine is mainly from the reductive carboxylation of glutamine rather than oxidation, the palmitate isotopologues distribution further confirmed reduced glutamine's contribution to palmitate for *de novo* lipid synthesis in USP13-KD HeyA8 and CAOV3 cells (Fig. 7d; Supplementary Fig. 7b). Conversely, in the USP13-overexpressed SKOV3 cells, the glutamine's contribution to palmitate increased (Fig. 7d). Consistent with these results, glutamine's percentage contribution to acetyl-CoA using isotopomer spectral analysis in the USP13-KD and USP13-overexpressed cancer cells is decreased and increased, respectively (Fig. 7e). Acetyl-CoA can also be converted from glucose-derived citrate by glucose oxidation. The palmitate isotopologue distribution shows that USP13-KD will also decrease glucose's contribution to *de novo* lipid synthesis in CAOV3 and HeyA8 (Supplementary Fig. 7c,d). Our results indicate that there is a decrease in total *de novo* synthesis of palmitate in USP13-KD cells, as well as diminished percentage contribution from glutamine reductive carboxylation and glucose oxidation to lipids (Supplementary Fig. 7e–g). Similar results were discovered for stearate in CAOV3 and HeyA8 cells under USP13-KD (Supplementary Fig. 7h–k).

**USP13 inhibition suppresses *de novo* lipid synthesis.** The loss of OGDH and ACLY in the USP13-KD cells may explain this reduced *de novo* lipogenesis. As proven above, KD of OGDH will significantly decrease the citrate pool in both CAOV3 and HeyA8, where citrate is the major precursor for lipogenic acetyl-CoA. USP13-KD also significantly decreased ACLY activity, while USP13 overexpression enhanced ACLY activity in the ACLY activity assay (Supplementary Fig. 7l). We observed that lipid

**Figure 6 | Depletion of USP13 inhibits glutaminolysis and induces mitochondrial dysfunction.** (**a**) Schematic of carbon atom transitions using [U-$^{13}C$] glutamine shows glutamine-driven oxidative metabolism into TCA cycle metabolites. (**b**) USP13 knockdown increases α-KG levels in CAOV3 and HeyA8 cells. Ctrl, shControl; KD-1, shUSP13-1; KD-2, shUSP13-2 ($n = 3$, each group). α-KG is transaminated with the generation of pyruvate that is utilized to convert a nearly colourless probe to both colour ($\lambda_{max} = 570$ nm) and fluorescence (Ex/Em = 535/587 nm). Unpaired *t*-test was used for statistical analysis. (**c**) Contribution of glutamine to TCA metabolites through oxidative metabolism and glutamate pool in control and USP13 knockdown HeyA8 and CAOV3 cells. Mass isotopologues distributions (MIDs, % of pool) of labelled metabolites are shown. (**d**) USP13 knockdown decreased NADH/NAD$^+$ ratio. (**e,f**) USP13 knockdown inhibits OCR in CAOV3 (**e**) and HeyA8 (**f**) cells. Basal OCR is a measure of OXPHOS activity. Oligomycin, FCCP and antimycin were used to assess mitochondrial functional state in the cells through maximal mitochondrial capacities ($_{Max}$) and respiratory control ratio (RCR). (**g**) USP13 knockdown decreases ATP levels in CAOV3 and HeyA8 cells. (**h**) Effect of glutamine on mitochondrial respiration in control and USP13 knockdown CAOV3 cells. (**i**) Schematic of carbon atom transitions using [U-$^{13}C$] glucose. (**j**) Relative level of M2 fumarate, malate and citrate that are derived from [U-$^{13}C$] glucose (Glc) in control and USP13 knockdown CAOV3 and HeyA8 cells. (**k**) Relative level of M4 fumarate, malate and citrate that are derived from [U-$^{13}C$] glutamine (Gln) in control, OGDH knockdown, ACLY knockdown and OGDH/ACLY double knockdown HeyA8 cells. (**l**) Supplementing cells with dimethyl-aspartate, dimethyl-malate partially rescues proliferation of USP13 knockdown cells. In this figure, error bars represent ± s.e.m. and $n \geq 6$. *$P < 0.05$, **$P < 0.01$, ***$P < 0.001$.

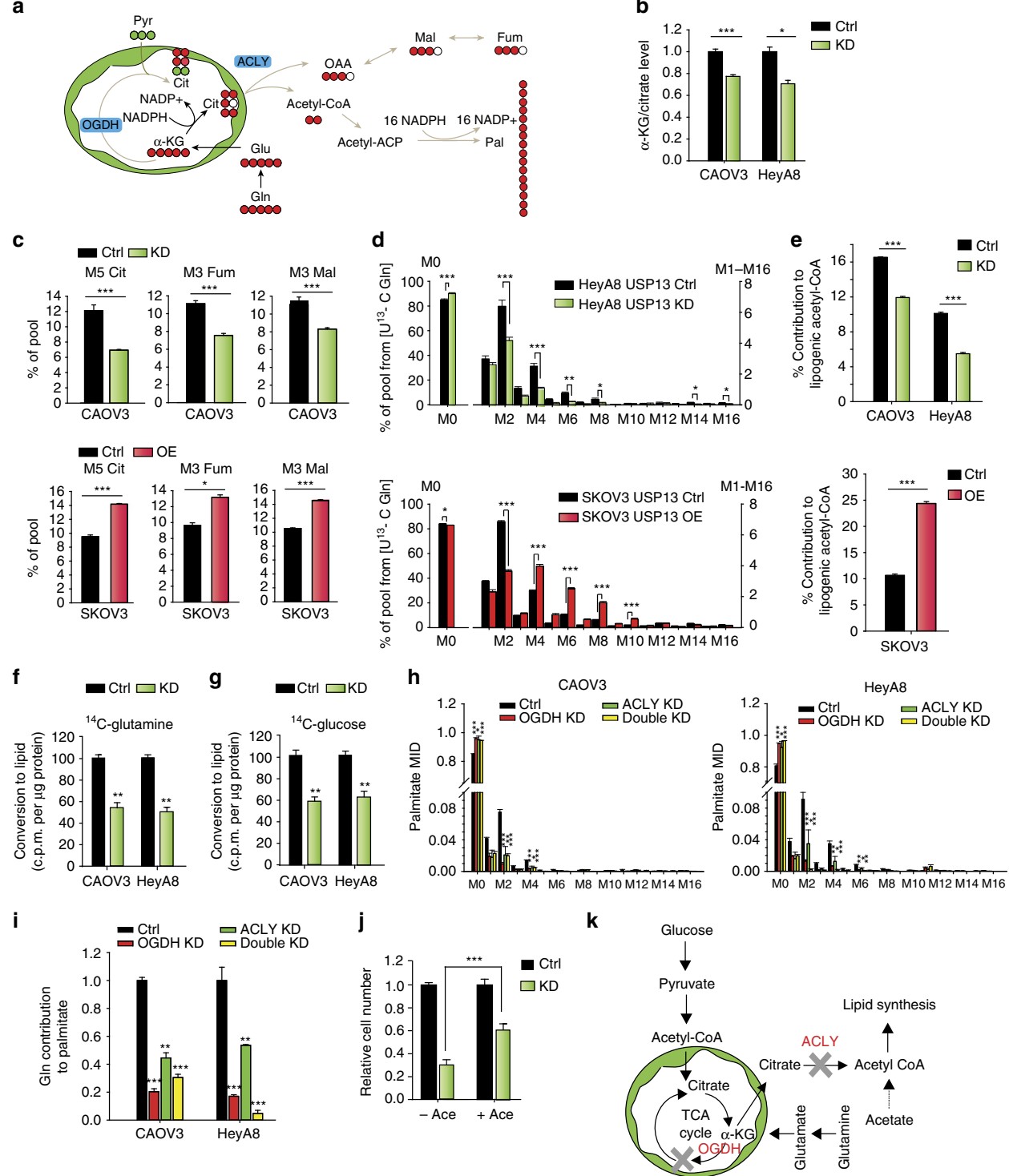

**Figure 7 | Depletion of USP13 inhibits ACLY in glutamine reductive carboxylation and lipid synthesis.** (**a**) Schematic of carbon atom transitions using [U-$^{13}$C] glutamine shows glutamine-driven reductive metabolism into palmitate synthesis. (**b**) Ratio of intracellular α-KG over citrate levels. (**c**) Contribution of [U-$^{13}$C] glutamine to TCA metabolites through reductive metabolism in control and USP13 knockdown CAOV3 cells, and in control and USP13-OE SKOV3 cells. Mass isotopologue distributions (MIDs; % of pool) of M5 citrate, M3 fumarate and M3 malate are shown. (**d**) Labelling of palmitate extracts from control and USP13-altered (knockdown or overexpressed) HeyA8 and SKOV3 cells. (**e**) Isotopologue spectral analysis (ISA) of glutamine's contribution to lipid synthesis. ISA shows that USP13 knockdown decreases glutamine's contribution to intracellular acetyl-CoA through reductive carboxylation. (**f,g**) USP13 knockdown decreases the incorporation of two major acetyl-CoA precursors, glutamine (**f**) and glucose (**g**), into lipid. (**h**) Labelling of palmitate from control, OGDH knockdown, ACLY knockdown and OGDH/ACLY double knockdown CAOV3 and HeyA8 cells cultured for 72 h in medium containing U-$^{13}$C Gln. MID from U-$^{13}$C$_5$ Gln was measured. (**i**) Relative contribution of glutamine to *de novo* synthesized fatty acids in total fatty acids pool in CAOV3 and HeyA8 cells. (**j**) Supplementing cells with sodium acetate partially rescues proliferation of USP13 knockdown cells. (**k**) Schematic shows that USP13 knockdown inhibits OGDH and ACLY in OVCA cell metabolism. In this figure, error bars represent ± s.e.m. and $n \geq 6$. *$P < 0.05$, **$P < 0.01$, ***$P < 0.001$.

synthesis from both glucose and glutamine was significantly reduced when USP13 was knocked down (Fig. 7f,g). To further clarify the effects of OGDH and ACLY on *de novo* fatty acids synthesis in OVCA, we used [U-$^{13}$C$_5$] glutamine or [U-$^{13}$C$_6$] glucose to culture cancer cells for 72 h. Our results show that both OGDH KD and ACLY KD decrease the conversion of glutamine and glucose in palmitate (Fig. 7h; Supplementary Fig. 8a,b). Isotopomer spectral analysis indicates that OGDH KD and ACLY KD decreases the fraction of *de novo* synthesized palmitate in total cellular palmitate pool as well as the percentage contribution from glutamine reductive carboxylation and glucose oxidation (Fig. 7i; Supplementary Fig. 8c,d). Similar results were observed for the stearate in both CAOV3 and HeyA8 cells (Supplementary Fig. 8e–h).

Clearly, both OGDH and ACLY are essential to maintain cancer cell proliferation by maintaining lipogenic acetyl-CoA pools. However, besides the conversion of citrate into acetyl-CoA carried by ACLY, the acetate in media could also supply acetyl-CoA for lipogenesis and proliferation[47]. As expected, supplementing acetate in culture media significantly rescued proliferation of the USP13-KD cells (Fig. 7j). These data suggest that USP13-KD induces mitochondrial dysfunction and lipogenic dysfunction by reducing OGDH and ACLY activity, thereby reducing glutamine's reductive carboxylation and glucose's oxidation for lipid synthesis (Fig. 7k). Targeting USP13 may result in synthetically lethal metabolic targets, which not only block nutrient's entry into TCA, but also reduce *de novo* lipogenesis.

**USP13 depletion inhibits ovarian tumour progression.** CAOV3 cells expressing Dox-inducible control or USP13 shRNA (>80% KD; Supplementary Fig. 9a) were used to establish xenograft tumour models in nonobese diabetic/severe combined immuno-deficiency (NOD/SCID) mice. Administration of mice with Dox (2.0 mg ml$^{-1}$) in drinking water markedly decreased the growth of xenograft tumours derived from the USP13-KD CAOV3 cells (Fig. 8a, right panel). However, there was no substantial difference between Dox-treated and -untreated tumours derived from control CAVO3 cells (Fig. 8a, left panel). Examination of end-point tumour burden in Dox-treated or -untreated groups ($n = 10$ per group) demonstrated that depletion of USP13 resulted in profound decreases in tumour weight (78.5% reduction, $P < 0.001$) and number of nodules (68.7% reduction, $P < 0.001$; Fig. 8b). Moreover, lower metastatic activity of the USP13-depleted ovarian tumours was observed in common metastatic sites (mesentery, omentum, diaphragm, peritoneal wall, perihepatic site, pelvic and kidney) of HGSC (Fig. 8c; Supplementary Fig. 9b). Protein levels of USP13, ACLY and OGDH were simultaneously reduced in the Dox-treated tumours (Fig. 8d). Notably, we observed that KD of USP13 had relatively less effect on the SKOV3-derived tumours regarding tumour weight (48.7% reduction, $P < 0.001$) and number of nodules (31.2% reduction, $P = 0.0036$; Supplementary Fig. 9c), suggesting that targeting USP13 may have therapeutic selectivity on the USP13-amplified OVCAs.

Both ACLY and OGDH are key metabolic enzymes that drive cancer cell proliferation. Inhibiting each or both of them significantly suppressed the proliferation of CAOV3 and HeyA8 cells *in vitro*, and inhibited the growth of ovarian tumours derived from CAOV3 and HeyA8 *in vivo* (Fig. 8e; Supplementary Fig. 10a–c). These results are consistent with our observation on reduced growth of the USP13-KD ovarian tumours (Fig. 2d). To further determine the essential role of ACLY and OGDH in USP13-induced ovarian tumourigenesis, the expression of ACLY and OGDH was restored by exogenous overexpression in OVCA

cells with USP13-KD (Supplementary Fig. 10d). Overexpression of either or both of ACLY and OGDH markedly rescued the proliferation of CAOV3 and HeyA8 cells even when USP13 was knocked down. CAOV3- and HeyA8-derived ovarian tumour growth was also greatly rescued by ACLY and OGDH overexpression (Fig. 8f; Supplementary Fig. 10e,f).

Next, we determined whether inhibition of USP13 sensitizes ovarian tumours to the treatment of the pan-AKT inhibitor MK-2206 *in vivo*. Mice bearing CAOV3-derived tumours expressing Dox-inducible USP13 shRNA were randomly divided into four groups (Dox-untreated, Dox-treated, Dox-untreated + MK-2206 and Dox-treated + MK-2206). One week after the Dox treatment, MK-2206 was administered by oral gavage three times per week (Fig. 9a). In comparison with vehicle control, MK-2206 treatment modestly inhibited tumour growth, but tumours were significantly smaller in the mice co-treated with MK-2206 and Dox (Fig. 9b). Dox treatment alone appeared to have a greater tumour inhibitory effect than MK-2206 treatment. We confirmed the Dox-inducible KD of USP13 and inhibition of AKT signalling *in vivo* using western blot (Fig. 9c) and IHC analyses (Fig. 9d). Proliferation index of the resected tumours was assessed (Fig. 9e, right panel). Treatment of MK-2206 significantly inhibited the activity of AKT (indicated by pAKT levels; Fig. 9e, left and middle panels) in both control and USP13-KD tumours. However, the tumours co-treated with Dox and MK-2206 had a markedly lower percentage of Ki-67-positive cells compared with the untreated or MK-2206-treated tumours.

## Discussion

Metabolic rewiring in cancer cells may render them highly dependent on specific metabolic enzymes or processes, which can be exploited for cancer therapy[48]. However, there continue to be challenges to target cancer metabolism for treatment in understanding which metabolic pathways are altered in cancer[2]. With the availability of multi-level databases in cancer genomics, transcriptomics and proteomics on the horizon for clinical care, a starting point now is to identify genomic events that frequently occur and play causal roles in a particular type of cancer. In this study, we identified frequent copy-number gain of the *USP13* gene in OVCA and found two USP13 deubiquitination targets that determine glutaminolysis, glucose oxidation, mitochondrial respiration and lipid synthesis. Our findings suggest that USP13 amplification is likely an important driver in ovarian tumour progression. Amplification of USP13 allows OVCA cells to rely on glutamine anaplerosis to replenish the TCA cycle with metabolic intermediates, and USP13 knocked down represses mitochondrial function. The biogenesis of mitochondria and its integration of metabolism and cellular signalling in cancer cells have been proven to play an important role in cancer initiation, progression and metastasis[4,26,49,50]. Moreover, increased level of USP13 promotes core metabolic pathways to generate ATP, reducing equivalents and main precursors for lipid biosynthesis. Our study provides a potential therapeutic strategy in which targeting USP13 blocks biosynthesis of metabolic intermediates and lipids thereby simultaneously inducing energy stress and cell death.

The function of USP13 in tumourigenesis has been controversial from previous studies. A recent study highlighted an important function of USP13 in inducing Beclin-1-mediated autophagy[51]. USP13 is a component of the p53 regulatory loop by its interaction with USP10, a positive regulator of p53. Zhao *et al.*[38] demonstrated that USP13 deubiquitinates and stabilizes microphthalmia-associated transcription factor, an essential modulator for melanoma growth, suggesting that USP13 may be a viable therapeutic target for melanoma. However, USP13 also appears to stabilize PTEN, at least in breast cancer, implicating its

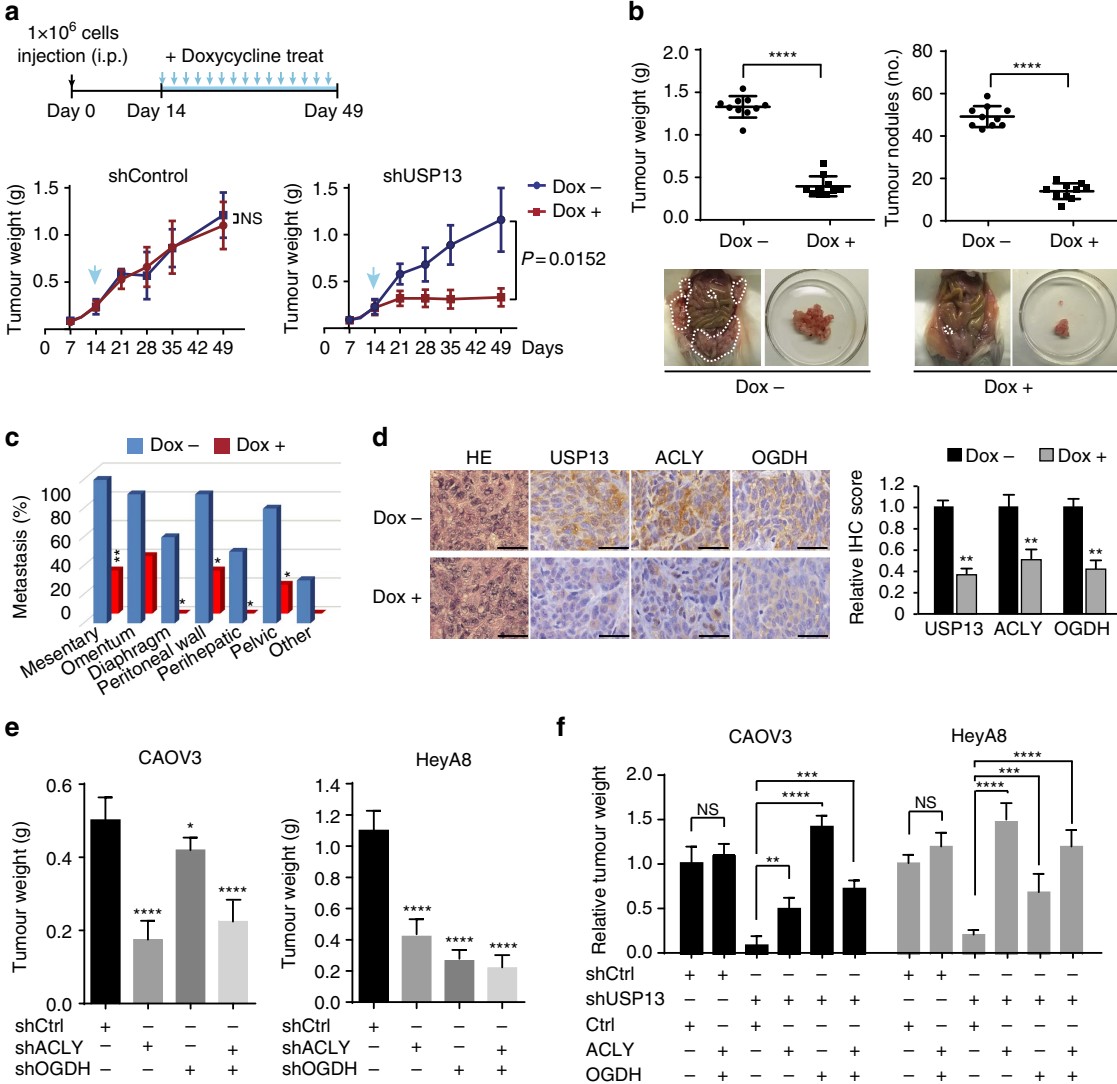

**Figure 8 | Inhibiting USP13 suppresses ovarian tumour growth and metastasis *in vivo*. (a)** Suppression of USP13 significantly inhibits ovarian tumour growth. CAOV3 cells expressing Dox-inducible control or USP13 shRNA were injected intraperitoneally into female NOD/SCID mice. Tumour growth was monitored every 7 days ($n = 5$ per time point). Dox treatment (red line) was initiated 14 days after injection. **(b)** Tumour weights and numbers of tumour nodules were measured in the CAOV3-derived tumours expressing Dox-inducible USP13 shRNA ($n = 10$ per group) 7 weeks after transplantation. Representative images of mice bearing Dox-treated or -untreated tumours are shown. **(c)** Metastatic sites and frequency of ovarian tumours derived from control and USP13 knockdown CAOV3 cells. Other sites include the kidney and liver. Significant difference in the metastatic patterns of these two groups was compared by Fisher's exact test. **(d)** IHC-staining analysis of USP13, ACLY and OGDH in Dox-treated or -untreated ovarian tumours. Scale bars, 50 μm. The graph shows relative signal intensity scores of USP13, ACLY or OGDH. **(e)** Knockdown of ACLY or OGDH significantly inhibits ovarian tumour growth *in vivo*. ACLY or OGDH was stably knocked down in CAOV3 or HeyA8 cells. The non-silencing scramble shRNA was used as a control. Cells ($1 \times 10^6$) expressing control scramble shRNA, shACLY or/and shOGDH were intraperitoneally injected into female NOD/SCID mice (seven mice in each group). Tumour weights were measured at 4 weeks after inoculation. **(f)** Overexpression of ACLY and/or OGDH rescued the growth of USP13 knockdown ovarian tumours *in vivo*. Relative tumour weights compared with control groups were shown. ACLY or/and OGDH were stably overexpressed in CAOV3 and HeyA8 cells expressing shControl or shUSP13. Cells ($1 \times 10^6$) of each group were intraperitoneally injected into female NOD/SCID mice (seven mice in each group). Tumour weights were measured 4 weeks after inoculation. Ctrl, empty expression vector. In this figure, unpaired *t*-test (two-tailed) was used for statistical analysis. Error bars represent ± s.d. *$P < 0.05$, **$P < 0.01$, ***$P < 0.001$, ****$P < 0.0001$. NS, nonsignificant.

tumour suppressive role[41]. Our analysis of cancer genomics reveals that the *USP13* gene is frequently amplified in human OVCA, but not in breast and colorectal cancers, suggesting that the role of USP13 in oncogenesis is context-dependent.

The PI3K/AKT/mTOR pathway is frequently altered in OVCA[34]. However, despite obvious biological rationale and positive results in preclinical models, clinical trials of first-generation mTOR inhibitors have demonstrated negative results in OVCA treatment[52]. Currently, a number of novel agents targeting mTORC1/mTORC2, AKT and PI3K are being

investigated for clinical intervention. Given the complexity and redundancy of PI3K/AKT signalling network in OVCA[33,35], it is crucial that efforts should be made to uncover potential resistance mechanisms. Our study shows that *USP13* is co-amplified with *PIK3CA* in the 3q26.3 amplicon in HGSC. We found that inhibiting USP13 sensitized OVCA cells to the treatment of AKT inhibitor, suggesting a synergistic role of USP13 in the PI3K/AKT-induced tumourigenesis. Amplification of USP13 may be a part of the mechanism for intrinsic resistance of PI3K/AKT inhibitors in the treatment of OVCA. The present study suggests

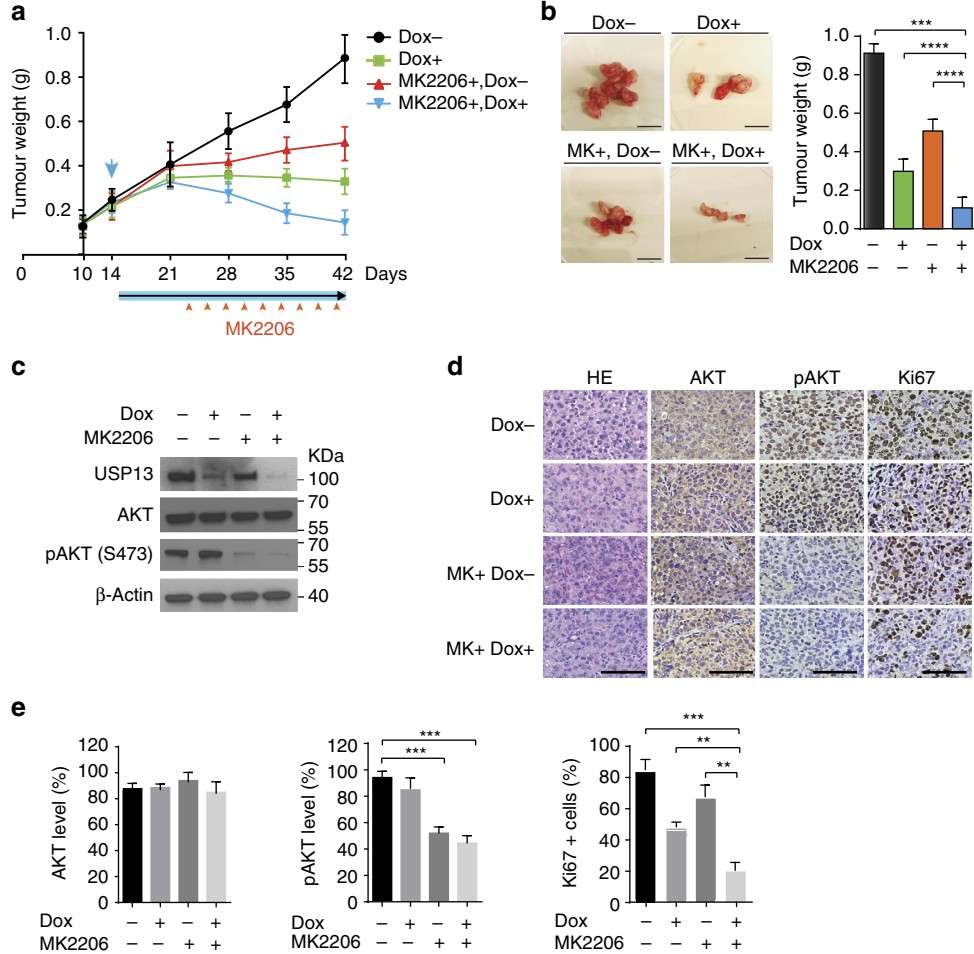

**Figure 9 | Knockdown of USP13 sensitizes ovarian tumours to the treatment of the AKT inhibitor.** (a) Effect of MK-2206 and USP13 knockdown on tumour growth *in vivo*. CAOV3 cells expressing Dox-induced USP13 shRNAs ($1 \times 10^6$) were injected intraperitoneally into female NOD/SCID mice ($n = 5$). Fourteen days after cell injection, Dox ($2\,\mathrm{mg\,ml}^{-1}$, 5% sucrose) treatment was initiated and 1 week later, MK-2206 at $120\,\mathrm{mg\,kg}^{-1}$ was administered by oral gavage (three times a week for 3 weeks). (b) Representative end-point tumour images (left) and tumour weights (right) in the four experimental groups ($n = 10$). Scale bars, 1 cm. (c) Efficiency of USP13 knockdown by Dox and inhibition of AKT signalling by MK-2206 *in vivo*. Tumours were collected 6 h after MK-2206 treatment and analysed for the levels of pAKT (S473), total AKT and USP13 by western blot. (d) Expression of total AKT, phospho AKT (S473) and Ki-67 in the above tumours was assessed by IHC staining. HE, haematoxylin and eosin staining. Scale bars, 100 μm. (e) Percentages of total AKT, Phospho-AKT1 (S473) and Ki-67-positive cells in each treatment group were compared. In this figure, unpaired *t*-test (two-tailed) was used for statistical analysis. Error bars represent ± s.d. \*\**P*<0.01, \*\*\**P*<0.001, \*\*\*\**P*<0.0001.

that combined treatment of PI3K/AKT inhibitor with USP13 inhibitor is a promising targeting strategy to overcome the resistance of PI3K/AKT inhibitors.

## Methods

**Tissue culture and tissues samples.** HEK293T, CAOV3, OVCAR3, HeyA8 and SKOV3 cell lines were purchased from the American Type Culture Collection. OAW28 cell line was purchased from Sigma. HEK293T cells were cultured in low-glucose DMEM with 10% fetal bovine serum at 37 °C in 5% $CO_2$. CAOV3, OWA28, IGROV1, OVCAR3, OVCAR8, HeyA8 and SKOV3 cell lines were grown in RPMI medium with 15% fetal bovine serum at 37 °C in 5% $CO_2$. All the cell lines were confirmed to have no contamination of mycoplasma using the Mycoplasma Detection kit (Lonza). Frozen tissue specimens of normal ovary and serous adenocarcinoma of ovary were received from Cooperative Human Tissue Network. All human subjects and human ovarian tissue samples were reviewed and approved by The University of Texas, MD Anderson Cancer Center Institutional Review Board (IRB). Informed consent was obtained from all subjects.

**Antibodies.** Anti-USP13 antibody (1:2,000, A302-762A) was purchased from Bethyl Laboratories. Anti-OGDH antibody (1:1,000, 15212-1-AP) was purchased from Proteintech. Anti-ACLY (1:1,000, ab61762) antibody was purchased from Abcam. Anti-Ki-67 (1:200, D3B5), anti-AKT (1:500, #2920), anti-phospho-AKT (1:500, Ser73; #4060) and anti-PTEN (1:2,000, #9552) antibodies were purchased

form Cell Signaling. Anti-actin (1:2,000, sc-1616), HRP-anti-goat IgG (1:3,000, #2020), HRP-anti-rabbit IgG (1:3,000, #2054) and HRP-anti-mouse IgG (1:3,000, #2055) antibodies were purchased from Santa Cruz. Antibodies against acetyl-histone H3 (Millipore #06-599), acetyl-histone H4 (1:1,000, Millipore #06-866), total histone H3 (1:1,000, Cell Signaling #9715) and total histone H4 (1:1,000, Cell Signaling #2592) were also used.

**Immunohistochemistry.** The tumours were fixed in 10% neutral buffered formalin and embedded in paraffin and 5-μm tissue sections were serially cut and mounted on slides. The sections were de-paraffinized in xylene, re-hydrated and boiled for 10 min in antigen retrieval buffer (BD1000 S-250, Borg Decloaker, BioCare Medical). After retrieval, the sections were washed with distilled water and endogenous peroxidase activity was blocked using 3% $H_2O_2$ in TBS for 15 min and then blocked with blocking solution (1% bovine serum albumin, 10% normal serum in 1× TBS). Samples were incubated with primary antibodies overnight at 4 °C, washed three times with TBST buffer and then incubated with biotinylated goat anti-rabbit or anti-mouse IgG (GR608H, BioCare Medical). A streptavidin–biotin peroxidase detection system was used according to the manufacturer's instructions (DAB Peroxidase Substrate kit or NovaRed Peroxidase Substrate kit, Vector Laboratory). Sections were counterstained with haematoxylin. Ovarian tumour tissue microarrays BC110118 and CJ2 were purchased from Biomax and Super Bio Chips, respectively. Signals of immunohistochemistry data in tumour cells were visually quantified using a scoring system from 1 to 3, multiplied intensity of signal and percentage of positive cells (signal: 0 = no signal, 1 = weak signal, 2 = intermediate signal and 3 = strong signal; percentage: 10–100%).

**Immunoblotting.** Total cell lysates were solubilized in lysis buffer (50 mM Tris, pH 7.5, 150 mM NaCl, 1 mM EDTA, 0.5% NP-40, 0.5% Triton X-100, 1 mM phenylmethylsulfonyl fluoride, 1 mM sodium fluoride, 5 mM sodium vanadate, 1 μg each of aprotinin, leupeptin and pepstatin per ml). Proteins were resolved by SDS–polyacrylamide gel electrophoresis gels and then proteins were transferred (Bio-Rad) to polyvinylidene difluoride membranes (Millipore). The membranes were blocked with 5% nonfat milk for 1 h at room temperature before incubation with indicated primary antibodies. Subsequently, membranes were washed and incubated for 1 h at room temperature with peroxidase-conjugated secondary antibodies (Santa Cruz Biotechnology). Following several washes, chemiluminescent images of immunodetected bands on the membranes were recorded on X-ray films using the enhanced chemiluminescence system (PerkinElmer) according to the manufacturer's instructions. Uncropped scans of the western blots are presented in the Supplementary Fig. 11.

**Generation of stable KD or overexpression cell lines.** Lentiviral pGIPZ vectors expressing non-silencing shRNA control or shUSP13, shACLY (#RHS4430-98903945) and shOGDH (#RHS4430-98896014) were obtained from the MD Anderson shRNA and ORFome Core Facility (originally from GE Dharmacon). To generate shRNA constructs against USP13, the following sequences are targeted: shUSP13-1: 5′-AAGGGAACATGTTGAAAGACAT-3′ and shUSP13-2: 5′-GCA TGTCGCAAGGCTGTGT-3′. To stably overexpress ACLY or OGDH, Precision LentiORF #OHS5897-101186358 (OGDH) and #OHS5898-219582244 (ACLY) were used from ORFome Core Facility (originally from GE Dharmacon). Cells were infected with lentiviruses in the presence of polybrene (8 μg ml⁻¹). To establish stable KD cell line, lentiviral shRNA-transduced cells were selected with puromycin (2–3 μg ml⁻¹) 48 h post infection and individual colonies were propagated and validated by western blotting (protein) and quantitative PCR with reverse transcription (messenger mRNA).

**Immunoprecipitation.** Cells were lysed on ice for 30 min in immunoprecipitation buffer (1% NP-40, 50 mM Tris-HCl, 500 mM NaCl and 5 mM EDTA) containing protease inhibitor cocktail. Cell lysates (700 μg) were incubated overnight with 3 μg of antibodies or normal IgG at 4 °C with rotary agitation. Protein A-sepharose beads were added to the lysates and incubated for additional 4 hours. Beads were washed three times with immunoprecipitation buffer and boiled for 10 min in 3% SDS sample buffer. Total cell lysates and immunoprecipitates were separated by SDS–polyacrylamide gel electrophoresis and analysed by western blotting[53].

**Sequence of primers for quantitative PCR with reverse transcription.**
Human USP13-F: 5′-GCGAAATCAGGCTATTCAGG-3′
Human USP13-R: 5′-TTGTAAATCACCCATCTTCCTTCC-3′
Human OGDH-F: 5′-ATCAGGGCATATCAGATACGAGG-3′
Human OGDH-R: 5′-TGATGAACATGAACTCCACCC-3′
Human ACLY-F: 5′-GGAAGTAGAGAAGATTACCACCT-3′
Human ACLY-R: 5′-CAAGATAGTGTCCAATGAACCC-3′

**In vivo deubiquitination assay.** 293T cells were transiently co-transfected with indicated plasmids. After 48 h, cells were treated with 5 μg ml⁻¹ MG132 (Sigma) for 6 h before being collected. Cells were lysed in denaturing buffer (6 M guanidine-HCl, 0.1 M Na₂HPO₄/NaH₂PO₄ and 10 mM imidazole). Cell lysates were then incubated with nickel beads for 3 h, washed and immunoblotted with indicated antibodies.

**In vitro deubiquitination assay.** For detection of ubiquitinated ACLY or OGDH in vitro, cell lysates were incubated with nickel beads and washed. The beads were incubated with purified USP13 in deubiquitination buffer (50 mM Tris-HCl (pH 7.5), 5 mM MgCl₂, 2 mM NaF and 0.6 mM dithiothreitol) for 1 h at 37 °C, and the bound proteins were eluted by boiling in 1 × Laemmli buffer and subjected to immunoblotting with indicated antibodies.

**Isotope-labelling analysis using GC–MS.** For metabolites extraction, cancer cells were seeded in six-well plates. Until the cells are 70–80% confluence, medium were replaced with isotope-labelling medium ([U-¹³C] glucose or [U-¹³C] glutamine). After 24 or 72 h, medium were aspirated, and cells were washed with ice-cold PBS to remove the remaining medium. Then, PBS was aspirated, 400 μl of ice-cold methanol was added into wells and another 400 μl of water containing 1 μg norvaline was added. Cells were scraped into 2-ml Eppendorf tubes. Another 800 μl of chloroform was added into the tubes. Samples were vortexed for 30 min at 4 °C, and then centrifuged for 10 min at a speed of 7,300 r.p.m. The top aqueous portion was collected for metabolites analysis, and bottom chloroform portion was collected for fatty acid analysis.

For derivatization, aqueous samples were dried at speed vac for 3 h to ensure their dryness, and dissolved in 30 μl of 2% methoxyamine hydrochloride in pyridine (Pierce). After 10-min sonication, samples were incubated at 37 °C for 2 h. Another 45 μl of MBTSTFA + 1% TBDMCS (Pierce) was added afterward, and

incubated at 55 °C for 1 h. Chloroform samples were dried and incubated at 60 °C with 75 μl methyl-8 reagent (Pierce) for 1 h. Samples were transferred into 150 μl of insert (Thermo Fish Scientific) for analysis.

Gas chromatography–mass spectrometry analysis was performed using an Agilent 6890 GC equipped with a 30-m Rtx-5 capillary column for metabolites samples connected to an Agilent 5975B MS. The following heating cycle was used for the GC oven: 100 °C for 3 min and a temperature increase to 300 °C with a rate of 5 °C min⁻¹. For fatty acids samples, a 30-m DB-35 MS capillary column was used with a heating cycle for the GC oven: 100 °C for 5 min, a temperature increase to 200 °C with a rate of 15 °C min⁻¹, and then a temperature increase rate at 5 °C min⁻¹ till 250 and 15 °C min⁻¹ to 300 °C. Metabolites abundance was analysed through the integral of peaks and normalized with internal standard norvaline.

Isotopologue spectral analysis was performed as previously described[54]. In brief, cells were incubated with ¹³C-labelled tracer (for example, U-¹³C6-glucose and U-¹³C5-glutamine) for 72 h to produce ¹³C-labelled fatty acids. The rate of de novo lipid synthesis (represented by the parameter g(t)) and fraction of acetyl-CoA pool (represented as parameter D) is calculated according to the mass isotopologues distribution of fatty acids. Through minimizing error between mass isotopologues distribution of fatty acids measured through gas chromatography–mass spectrometry and mass isotopologues distribution generated from binomial functions, the parameter D and g(t) were estimated.

**Lipid synthesis assay.** Cells were incubated in the medium containing D-[6-¹⁴C] glucose (5 μCi ml⁻¹, PerkinElmer) or [U-¹⁴C]-glutamine (10 μCi ml⁻¹, Cambridge Isotope Labs) for 8 h. Cells were washed three times in PBS. Cellular lipid was extracted by a modified Bligh Dyer method[55]. In brief, cells were lysed in a 0.5% Triton X-100 solution. Two millilitre of methanol, 1 ml of chloroform, and 1 ml of distilled water were added. After centrifugation, the chloroform phase was dried and resuspended in chloroform. The signals were counted in ScinitiVerse (Fisher) on a Beckman LS6500 scintillation counter. Assay was performed in triplicate, and the results were expressed as percentage change in c.p.m. (counts per minute) per μg protein.

**Measurements of OCR.** Mitochondrial OCR was measured by XF24 Analyzer (Seahorse Biosciences). Cells were seeded in 24-well seahorse plates and incubated at 37 °C with 5% CO₂ overnight. Medium was replaced with 700 μl medium free of serum and sodium bicarbonate. Plates were then incubated in CO₂-free incubator for 1 h before placing in an analyzer. The OCRs were measured with procedure of 3-min mixing, 2-min waiting and 3-min measuring. Oligomycin, FCCP and antimycin were injected through port A, B and C, respectively, to calculate mitochondrial function under different stress. Data were normalized with protein levels.

**NADH/NAD + assay.** Cells were seeded and cultured in a six-well plate overnight until 80% confluence. Mediums were replaced with fresh complete medium and cells were collected with 500 μl of NADH extraction buffer (20 mM nicotinamide, 20 mM NaHCO₃ and 100 mM Na₂CO₃, pH 8.3). The cells were lysed with two freeze/thaw cycles. NADH standard was serially diluted from 1 mM NADH stock solution. NADH cycling buffer (100 mM Tris-HCl solution at pH 8.5, 0.5 mM MTT or thiazolyl blue, 2 mM phenazine ethosulfate, 5 mM EDTA and 0.15 mg ml⁻¹ ADH) and NADH substrate (4% EtOH and 96% distilled water) were prepared. A volume of 160 μl of cycling buffer was added into 96-well plates, and then 20 μl of NADH standard or cell extracts were added with 20S shake on plate reader. A volume of 20 μl of NADH substrate was then added. Absorbance values were read at 570 nm for 30 min under kinetic mode.

**Ovarian tumour xenograft mouse model.** Female NOD/SCID mice (NOD.CB17-Prkdcscid/J, 6–8 weeks old) were purchased from Jackson Laboratories. All studies were approved and supervised by the Institutional Animal Care and Use Committee at the MD Anderson Cancer Center. To determine the effect of USP13, ACLY or OGDH KD on ovarian tumour growth, OVCA cells (1 × 10⁶) stably expressing control or USP13, ACLY or OGDH shRNA in 100 μl HBSS Hank's solution were injected intraperitoneally into mice after they were anaesthetized. In control or the USP13-KD group, the mice were randomly divided into non-treated and treated subgroups (n = 5 per time point, 30 mice per subgroup). Two weeks after cell injection, normal drinking water was replaced with 5% sucrose with 2 mg ml⁻¹ Dox and changed every day. The primary tumours were collected and weighed, and the number of tumours nodules was measured every week. For endpoint study, mice were randomly divided into non-treated and treated groups (n = 10 mice per group), and the mice were killed before they met the institutional euthanasia criteria for tumour burden and overall health condition.

To determine whether USP13-KD sensitizes ovarian tumour cells to the treatment of MK-2206, CAOV3 cells (1 × 10⁶) stably expressing Dox-inducible USP13 shRNA were injected intraperitoneally into female NOD/SCID mice. Ten days after cell injection, mice bearing tumours were randomly divided into four treatment groups (Dox-untreated, Dox-treated, Dox-untreated + 120 mg kg⁻¹ MK-2206 and Dox-treated + 120 mg kg⁻¹ MK-2206, 30 mice per group and 5 mice killed per time point in each group). MK-2206 (120 mg kg⁻¹ body weight)

was administered for 3 weeks (three times a week) by oral gavage. Animals were killed 6 h after the last dose and tumours were resected, fixed with 4% paraformaldehyde and paraffin-embedded for IHC staining.

**Data availability.** The TCGA data referenced during the study are available in a public repository from the cBioPortal for Cancer Genomics, TCGA website (http://www.cbioportal.org/). The authors declare that all the other data supporting the findings of this study are available within the article and its Supplementary Information files and from the corresponding authors on reasonable request.

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

## Acknowledgements
This work was supported by grants to X.L. from the National Institutes of Health (NIH; CA185742 and CA203737), to C.Ha. from the NIH (CA197487) and to X.Z. from Cancer Prevention & Research Institute of Texas (CPRIT; RP150093).

## Author contributions
C.Ha., D.N. and X.L. designed the experiments and wrote the paper. C.Ha., L.Y. and G.J. conducted *in vitro* and *in vivo* OVCA cell studies. L.Y., J.B. and A.A. performed metabolic experiments. H.H.C., Y.Liu, Y.Li and X.Z. analysed protein ubiquitination. C.Ha., G.W., J.L. and C.Hu. conducted bioinformatic and biostatistical analyses.

## Additional information

**Competing financial interests:** The authors declare no competing financial interests.

