## [Peer Review File · Nature Communications]

Reviewers' comments:

Reviewer #1 (Expert in Ubiquitination and cancer)
(Remarks to the Author):

The manuscript by Han et al. studied the role of de-ubiquitinase USP13 in ovarian cancer metabolic regulation and tumor progression. In this paper, the authors found that the USP13 gene is frequently amplified in human OVCA, and its overexpression is significantly associated with poor clinical outcome. They further demonstrated that USP13 knockdown impaired ovarian cancer growth in vitro and in vivo. Mechanistically, they showed that USP13 specifically deubiquitinates and thus up-regulates ATP citrate lyase (ACLY) and oxoglutarate dehydrogenase (OGDH), two key enzymes that determine mitochondrial respiration, glutaminolysis, and fatty acid synthesis. Importantly, USP13 silencing sensitizes ovarian cancer cells to the treatment of PI3K/AKT inhibitor both in vitro and in vivo. Overall, these results provide important evidence for the oncogenic function of USP13 in OVCA. Also, the studies by Han et al. have not only revealed new targets for the de-ubiquitinase USP13, but also unraveled a novel link between USP13 and tumor metabolism regulation. Experiments presented in this manuscript are well controlled, and data quality in general is good and convincing. However, the authors need to address the critical issues as outlined below in order to further support their conclusion and for this manuscript to be considered for publication in Nature Communication.

1. Although the authors have provided strong evidence demonstrating the critical role of USP13 in OVCA cell proliferation and tumorigenesis, it is not firmly proven that in vivo oncogenic role of USP13 indeed acts through OGDH and ACLY regulation. More direct evidence is needed to support the authors' conclusion. To this end, it is necessary to show that suppression of USP13 deletion on tumor growth in vivo is reversed by re-expression of OGDH or ACLY in USP13 knock down cells.
2. In figure 1, the author showed that USP13 gene is amplified in several type of human cancer, especially in OVCA. To further confirm this conclusion, they performed immunohistochemical (IHC) analysis by using a tumor tissue microarray. It will be more convincing if the authors could provide the Western blot results by comparing the expression level of USP13 in several pairs of OVCA cancer samples and their corresponding normal adjacent tissues.
3. The authors concluded that USP13 removes K48-linked ubiquitination of ACLY and OGDH by using ubiquitin K48R and K63R constructs. However, such conclusion is thus far premature. To make such claim, it is necessary to use HA-ubiquitin K48 only and K63 only constructs for the ubiquitination assay.
4. The effect of USP13 on deubiquitination of OGDH in Fig. 4f is marginal. The authors should repeat the experiment to present more convincing results.
5. USP13 is found amplified in ovarian cancer and predicts poor survival outcome. Does USP13 overexpression promote ovarian cancer development? If so, does such regulation depend on USP13 catalytic activity?
6. USP13 was found to stabilize PTEN expression by removing PTEN Ubiquitination (Nature Cell Biology 15, 1486-1494 (2013)). Does USP13 regulate PTEN and Akt activity in ovarian cancer cells?
7. To better understanding the role of USP13 in tumor metabolic regulation, the authors should provide more information about the role of ACLY and OGDH in cancer regulation in the introduction.
8. As regulation of USP13 on ACLY and OGDH is one of the major findings in this paper, the authors should move the supplementary data related to the expression level of these proteins and the positive correlation between USP13 and either ACLY or OGDH to the main figures.

Reviewer #2 (Expert in Cancer metabolism)
(Remarks to the Author):

The manuscript contributed by Han and colleagues identified that USP13 serves as a new molecular target in ovarian cancers. The authors performed a series of studies to reveal that USP13 upregulates ACLY and OGDH via direct interaction and deubiquitinating these proteins. Further, the authors perform stable isotope flux experiments using ¹³C-labeling glucose and glutamine to examine the flux of metabolites to lipids and mitochondria intermediates to reveal the effects of inhibition of USP13 on metabolism. The authors also showed that genetic inhibition of USP13 significantly suppressed ovarian tumor growth in mouse xenograft model. Overall, the study is an outstanding work. Experiments were well designed and executed. The writing is also excellent and clear.

A few points could further strengthen this study.

1. The acetyl-CoA derived from Citrate by ACLY also serves as a precursor for protein acetylation, thus, beyond lipid synthesis and glutamine metabolism, USP13 may also regulate protein acetylation via upregulation of ACLY. The authors could quickly check histone acetylation to examine whether USP13 is also involved in protein acetylation via regulation of ACLY. If the results are positive, the conclusion and discussion should include this result.
2. Figure 2, whether knockdown ACLY and OGDH could have similar effects on CAOV3 and HetA8 xenograft tumor growth as shUSP13? These experiments should be tested.
3. Figure 3, please clarify "His-ubi" and "No-NTA" in the legend.
4. Figure 5J and I, knockdown of USP13 enhanced the levels of ¹³C-labeled citrate and fumarate, whether these effects were caused by the reduction of ACLY or OGDH are not clear. shACLY, shOGDH and double knockdown could be applied to examine ¹³C-glucose flux. The increase of ¹³C-citrate and ¹³C-fumarate may be caused by the reduction of ACLY and its-mediated fatty acid synthesis flux, not caused by increasing or switching to glucose uptake. ¹⁴C-glucose could be used to check whether shUSP13 enhances glucose uptake.
5. Figure 7d, IHC staining for the decrease of USP13, ACLY, OGDH is not clear. Please consider to use Novo Red staining substrates to get better comparison of images.
6. Please consider to change all "KD of USP13" to shRNA "knockdown of USP13" in order to avoid the confusion with "kinase dead".

Reviewer #3 (Expert in Metabolomics techniques)
(Remarks to the Author):

The metabolite profiling and mass spectrometry aspects of this proposal seem to be technically sound; appropriate experimental descriptions are provided, as are reasonable diagrams for tracing heavy carbon through metabolism.

There are some questions about the interpretation of the data, though. Specifically, Figures 5 and 6 seem to be treated distinctly for each specifically addressing the impact of one gene whose regulation is changed by USP13 overexpression or knockdown, but the data are probably more coupled than is described by the authors. Starting with Figure 5b, the authors say that they expect higher AKG levels due to lower OGDH. That seems reasonable; one would then also assume there would be lower levels of, say malate and fumarate since they are after the OGDH step (with perhaps decreasing correlation the further down the cycle these metabolites are). However, that evidence is instead presented separately, in the discussion of ACLY. In that discussion, and based on Figure 6A, the authors seem to be suggesting that cytosolic malate and fumarate are the things to be measured, rather than mitochondrial levels. However, their sample prep does not entail a step to separate or isolate mitochondria, so the levels they measure are in the whole cell. Accordingly, to treat the lower malate and fumarate levels as independent evidence of ACLY impact and not, say, the TCA-related impact of

OGDH, seems questionable.

Additional explanation is also needed for the authors' seeming expectation that decreased contribution of glutamine-derived carbon in an OGDH-down condition is to be expected. Does OGDH not act on all of the substrate in the TCA cycle, not just the glutamine-derived fraction? Should a change in the TCA cycle not affect everything equally? Or is this coming specifically from the cofactor aspect of their discussion? Is OGDH somehow uniquely linked to glutaminolysis? Or perhaps is it coming from the ACLY-related changes? Again, the labeling data seem to support their factual description of fluxes, but the interpretation and expectations are unclear.

It seems like the authors are ultimately talking about a shift from glutaminolysis to use of glucose with USP13 knockdown, but are trying to mechanistically explain it via the action of a TCA enzyme. This needs to be explained better/more clearly.

And overall, the story seems to be saying that there is overexpression of USP13 in OVCA, which is causing deubiquitination and thus removal of ubiquitin that would otherwise target OGDH for degradation, and thus more OGDH, and thus more TCA cycle enzymes... this seems to be in contrast to the typical expectation that oxphos is not the dominant mode in cancer cells. This should be addressed more head-on and reconciled, perhaps in the discussion.

There is just a lot of interpretation here where this reviewer is not sure on how consistent it all is with each other and with what is already known. The mechanistic interpretations need to be much more concretely justified.

Reviewer #4 (Ovarian Cancer)
(Remarks to the Author):

Han and colleagues investigated the role of USP13 in the regulation of metabolism and tumorigenesis in high-grade serous ovarian cancer (HGSC). Analysis of TCGA data showed that USP13 is co-amplified with PIK3CA, and that over expression was associated with poor outcome. USP13 was shown to directly deubiquitinate ACLY and OGDH, and gene suppression in amplified/overexpressing cells reduced cell proliferation and tumor progression in vivo. Inhibition of USP13 was also shown to sensitize cells to the pan-AKT inhibitor MK-2206. The authors propose a therapeutic strategy for HGSC involving combined inhibition of USP13 and the PI3K/AKT pathway. The study was comprehensive and findings were clearly and logically presented.

Q1) USP13 protein levels were analyzed by IHC on TMA. In figure 1e, were the adjacent normal samples and HGSC samples presented matched? Y-axis is labeled as intensity but the values appear to be a combination of intensity and % cells stained. Authors should define 'a.u.' (arbitrary units?).

Q2) Authors should indicate if control constructs used in shRNA experiments were empty vector or scramble?

Q3) In Figure 2C, are the clonogenic assay images shown from KD-1 or KD-2? At what time point where colonies counted? Methods should be ordered in line with Results.

Q4) Has PIK3CA and/or pathway member protein expression been characterized in cell lines? Is PIK3CA amplification or PI3K/AKT pathway activation associated with MK2206 sensitivity? It is not clear why the authors chose to look at a pan-AKT inhibitor rather than a PI3K inhibitor in the first instance.

Q5) It should be noted that HeyA8, A2780, SKOV3 are not considered to be representative of HGSC (Domcke et al., 2015).

Q6) Error bars should be presented in Figure 7 c and N should be reported for 7d.

Q7) Were markers other than Ki67 quantitated for data presented in Figure 7h.

Q8) Can the authors conclude that the effect of USP13 suppression by shRNA KD and MK2206 treatment is synergistic rather than additive? Has synergy between MK2206 and USP13 inhibitors been assessed (e.g. using Chou-Talalay methodology)?

Response to Reviewers (Revised manuscript NCOMMS-16-03474-T)

Attached for consideration for publication as a Research Article in *Nature Communications* is our revised manuscript entitled, “**Amplification of USP13 drives ovarian cancer metabolism**”. We thank you for your helpful comments and critiques. In the revised manuscript, we believe that we have addressed the reviewers’ concerns arising from the initial manuscript. In particular, we analysed the expression levels of OGDH and ACLY in ovarian tumour samples and their correlation with USP13 levels. We also investigated their functional role in vivo using ovarian tumour models with altered levels (overexpression and knockdown) of OGDH and ACLY. Moreover, we also provided additional data and discussion addressing PIK3CA pathway activation in the tested cell lines and its correlation to MK2206 sensitivity. In addition, we have also performed a number of metabolic analyses and assays to further support the role of USP13 in ovarian cancer metabolism. By addressing these concerns, we have considerably strengthened the manuscript and hope you agree. Additional data has been added into the manuscript and almost all of the figures have been modified to include new results. I have responded to the review comments on a point-by-point basis as below:

Reviewer #1:

1. *“Overall, these results provide important evidence for the oncogenic function of USP13 in OVCA. Also, the studies by Han et al. have not only revealed new targets for the de-ubiquitinase USP13, but also unravelled a novel link between USP13 and tumor metabolism regulation. Experiments presented in this manuscript are well controlled, and data quality in general is good and convincing. However, the authors need to address the critical issues as outlined below in order to further support their conclusion and for this manuscript to be considered for publication in Nature Communication.”*

We thank Reviewer#1 for his/her positive comment and helpful suggestions. In the revised manuscript, we include a significant number of additional data (Fig. 1f, 4f, 5, 8f and Supplementary Fig. S2, S5, S10d, S10e, S10f) to address the reviewer’s concerns and further support our conclusion.

2. *“Although the authors have provided strong evidence demonstrating the critical role of USP13 in OVCA cell proliferation and tumorigenesis, it is not firmly proven that in vivo oncogenic role of USP13 indeed acts through OGDH and ACLY regulation. More direct evidence is needed to support the authors' conclusion. To this end, it is necessary to show that suppression of USP13*

deletion on tumor growth in vivo is reversed by re-expression of OGDH or ACLY in USP13 knock down cells.”

We completely agree with the reviewer and have performed the in vivo experiments as shown in Fig. 8f and Supplementary Fig. S10d, 10e, 10f. The results showed that overexpression of ACLY, OGDH or ACLY/OGDH significantly restored the growth of the USP13-knockdown ovarian tumours.

3. *“In figure 1, the author showed that USP13 gene is amplified in several type of human cancer, especially in OVCA. To further confirm this conclusion, they performed immunohistochemical (IHC) analysis by using a tumor tissue microarray. It will be more convincing if the authors could provide the Western blot results by comparing the expression level of USP13 in several pairs of OVCA cancer samples and their corresponding normal adjacent tissues.”*

As suggested, we measured the expression levels of USP13 in human ovarian tumours and their normal control tissues. The results showed that USP13 is expressed at markedly higher levels in the tumours than in the normal tissues ($p=0.0157$, shown in Fig. 1f and Supplementary Fig. S2).

4. *“The authors concluded that USP13 removes K48-linked ubiquitination of ACLY and OGDH by using ubiquitin K48R and K63R constructs. However, such conclusion is thus far premature. To make such claim, it is necessary to use HA-ubiquitin K48 only and K63 only constructs for the ubiquitination assay”.*

We thank the reviewer for pointing out this concern. As suggested, we performed in vivo deubiquitination assays to confirm that USP13 deubiquitinates ACLY and OGDH at K48 (Supplementary Fig. S5). K48-only and K63-only ubiquitin constructs were used to label the ubiquitinated form of USP13. Neither of ACLY and OGDH is poly-ubiquitinated by K63-ubiquitin. However, they can both be ubiquitinated by K48-only ubiquitin and this type of ubiquitination can be removed by USP13. The results further support that USP13 removes K48-ubiquitination of ACLY and OGDH.

5. *“The effect of USP13 on deubiquitination of OGDH in Fig. 4f is marginal. The authors should repeat the experiment to present more convincing results.”*

In vivo deubiquitination assay is technically challenging. We agree with the reviewer and have optimized the experiment by adjusting the relative expression levels of USP13 and OGDH in cells. The results now provide solid evidence that USP13 deubiquitinates K48-ubiquitinated OGDH (Fig. 4f).

6. *“USP13 is found amplified in ovarian cancer and predicts poor survival outcome. Does USP13 overexpression promote ovarian cancer development? If so, does such regulation depend on USP13 catalytic activity?”*

We believe that USP13 is a critical player in the 3q26 amplicon that promotes ovarian cancer metabolism. Knockdown of USP13 dramatically inhibits ovarian tumour growth as shown in Figs 2, 8, and 9. However, it is unknown whether USP13 amplification is sufficient to promote ovarian tumorigenesis by its own. In my laboratory, we are planning to generate a tissue-specific USP13 (wildtype and DUB-inactive form) transgenic mouse model, which will answer this critical question. Based on our cell-based studies, the function of USP13 in regulation of cancer cell metabolism is dependent on the DUB (deubiquitination) activity USP13. The C345A mutant of USP13 loses its activity on ACLY and OGDH (Fig. 3, 4).

6. *“USP13 was found to stabilize PTEN expression by removing PTEN Ubiquitination (Nature Cell Biology 15, 1486-1494 (2013). Does USP13 regulate PTEN and Akt activity in ovarian cancer cells?”*

We analysed the protein levels of PTEN in control and USP13-KD CAOV3 and HeyA8 cells. No notable changes on the PTEN levels were observed in the USP13-knockdown cells (Supplementary Fig. 4g), suggesting that PTEN is not a key target of USP13 at least in ovarian cancer cells and that USP13 may act in a context-dependent manner.

7. *“To better understanding the role of USP13 in tumor metabolic regulation, the authors should provide more information about the role of ACLY and OGDH in cancer regulation in the introduction.”*

As suggested, we have provided more information about the role of ACLY and OGDH in cancer regulation in the Introduction.

8. *“As regulation of USP13 on ACLY and OGDH is one of the major findings in this paper, the authors should move the supplementary data related to the expression level of these proteins and the positive correlation between USP13 and either ACLY or OGDH to the main figures.”*

We agree with reviewer and have moved the results related to the expression levels of ACLY, OGDH and USP13 from supplementary figures to Fig. 5.

Reviewer #2:

1. *“Overall, the study is an outstanding work. Experiments were well designed and executed. The writing is also excellent and clear. A few points could further strengthen this study..”*

We thank Reviewer #2 for his/her positive comments in our studies. In the revised manuscript, we have already addressed each point raised out in the review comments.

2. *“The acetyl-CoA derived from Citrate by ACLY also serves as a precursor for protein acetylation, thus, beyond lipid synthesis and glutamine metabolism, USP13 may also regulates protein acetylation via upregulation of ACLY. The authors could quickly check histone acetylation to examine whether USP13 is also involved into protein acetylation via regulation of ACLY. If the results are positive, the conclusion and discussion should include this result. ”*

We fully agree with the reviewer that USP13 may regulate histone acetylation and global gene transcription through ACLY. As suggested, we have performed experiments to check the levels of total and acetylated H3 and H4 in the USP13-amplified CAOV3 cells when USP13 was knocked down (Supplementary Fig. 4f). Our results showed that USP13 knockdown has no notable changes on the acetylation levels of H3 and H4. However, we do not exclude the possibility that particular histone acetylation sites or other histones may be regulated by USP13. We have included this result in the manuscript.

3. *“Figure 2, whether knockdown ACLY and OGDH could have similar effects on CAOV3 and HetA8 xenograft tumor growth as shUSP13? These experiments should be tested.”*

As suggested, we investigated whether knockdown of ACLY and OGDH had similar effects on CAOv3 and HeyA8 xenograft tumour growth (Fig. 8e and Supplementary Fig. S10a, S10b, S10c). The results showed that ovarian tumour growth was significantly inhibited by knockdown of both ACLY and OGDH.

4. *“Figure 3, please clarify “His-ubi” and “No-NTA” in the legend.”*

We have clarified these abbreviations in the figure legend.

5. *“Figure 5J and I, knockdown of USP13 enhanced the levels of 13C-labeled citrate and fumarate, whether these effects were caused by the reduction of ACLY or OGDH are not clear. shACLY, shOGDH and double knockdown could be applied to examine 13C-glucose flux. The increase of 13C-citrate and 13C-fumarate may be caused by the reduction of ACLY and its-mediated fatty acid synthesis flux, not caused by increasing or switching to glucose uptake. 14C-glucose could be used to check whether shUSP13 enhances glucose uptake.”*

We thank the reviewer for pointing this out. In our revised manuscript, we knocked down the gene expression of OGDH, ACLY individually or OGDH and ACLY together. Interestingly, we found that OGDH KD and double KD have profound effects on reducing both glutamine oxidation and glucose oxidation (Fig. 6k, Supplementary Fig. S6e, 6f). OGDH/ACLY double knockdown increased glycolysis in both CAOv3 and HeyA8, a similar observation as made in the USP13-knockdown cells (Supplementary Fig. S6c and S6d). However, ACLY KD alone did not have a significant effect on TCA cycle metabolism in both CAOv3 and HeyA8 cells (Fig. 6k, Supplementary Fig. S6e, S6f). This is because OGDH is essential for maintaining aspartate, a critical metabolite for cell proliferation. Indeed, we observed that supplementing cells with aspartate and malate partially rescued the proliferation of the USP13-KD CAOv3 cells (Fig. 6l). Therefore, we definitively prove that USP13 can regulate mitochondrial TCA cycle metabolism by upregulating OGDH expression level. Our results suggest that USP13 increased glycolysis by diverting glucose more towards lactate secretion. This was also evident from reduced glucose contribution towards TCA cycle metabolites.

6. *“Figure 7d, IHC staining for the decrease of USP13, ACLY, OGDH is not clear. Please consider to use Novo Red staining substrates to get better comparison of images.”*

We thank for the reviewer's suggestion. Nova Red staining was performed on tumour tissues and the protein levels of USP13, ACLY, and OGDH were analysed and quantified (Fig. 8d). The results showed that Dox-induced knockdown of USP13 significantly reduced the levels of ACLY and OGDH in the xenograft ovarian tumours.

7. *"Please consider to change all "KD of USP13" to shRNA "knockdown of USP13" in order to avoid the confusion with "kinase dead"."*

We have done it as suggested in the revised manuscript. Thank you for the suggestion.

Reviewer#3:

1. *"The metabolite profiling and mass spectrometry aspects of this proposal seem to be technically sound; appropriate experimental descriptions are provided, as are reasonable diagrams for tracing heavy carbon through metabolism."*

We thank the reviewer for his/her positive comments. In the revised manuscript, we have performed a number of additional experiments to address all the review comments to further support our conclusion.

2. *"There are some questions about the interpretation of the data, though.....Starting with Figure 5b, the authors say that they expect higher AKG levels due to lower OGDH. That seems reasonable; one would then also assume there would be lower levels of, say malate and fumarate since they are after the OGDH step (with perhaps decreasing correlation the further down the cycle these metabolites are). However, that evidence is instead presented separately, in the discussion of ACLY. In that discussion, and based on Figure 6A, the authors seem to be suggesting that cytosolic malate and fumarate are the things to be measured, rather than mitochondrial levels. However, their sample prep does not entail a step to separate or isolate mitochondria, so the levels they measure are in the whole cell. Accordingly, to treat the lower malate and fumarate levels as independent evidence of ACLY impact and not, say, the TCA-related impact of OGDH, seems questionable."*

We would like to clarify that we have divided the data obtained from isotope tracing using labelled glucose and glutamine for clarity. In Fig. 6 (original Fig.5), the results that show metabolic effects of USP13 through OGDH and ACLY are presented. In Fig. 7 (original Fig. 6),

effects of USP13 on de novo fatty acids synthesis via modulations of ACLY are shown. This was also necessary to streamline discussion on the effects of USP13 in regulating cell proliferation by maintaining TCA cycle metabolites and emphasize separately the role of USP13 in maintaining proliferation through lipogenesis. The use of labelled U- $^{13}\text{C}_5$ glutamine helps us in dissecting the role of glutamine driven oxidative metabolism (through OGDH) and glutamine driven reductive metabolism (through ACLY). When proliferating cells utilize glutamine by oxidative metabolism, they produce M4 fumarate, M4 malate, and further combine oxaloacetate with acetyl-CoA to form M4 citrate (obtained by condensing of labeled oxaloacetate obtained from glutamine and unlabeled acetyl CoA). Alternatively, cells under hypoxia or under mitochondrial electron transport inhibition have been reported to predominantly use reductive carboxylation of glutamine to produce α -KG through IDH1/2 and then to generate M5 citrate. M5 citrate is further catalyzed to M3 fumarate and M3 malate in this reductive glutamine metabolism. A significant increase in M5 citrate, M3 fumarate and M3 malate in cancer cells suggests that cancer cells rely critically on reductive glutamine metabolism when normal mitochondrial function is disrupted. Thus, use of metabolite isotope tracing allows us to dissect the USP13 effects through OGDH from USP13 effects through ACLY. In addition, we have added knockdown results of OGDH and ACLY that further clarifies the effects of individual enzymes.

3. "Additional explanation is also needed for the authors' seeming expectation that decreased contribution of glutamine-derived carbon in an OGDH-down condition is to be expected. Does OGDH not act on all of the substrate in the TCA cycle, not just the glutamine-derived fraction? Should a change in the TCA cycle not affect everything equally? Or is this coming specifically from the cofactor aspect of their discussion? Is OGDH somehow uniquely linked to glutaminolysis? Or perhaps is it coming from the ACLY-related changes? Again, the labeling data seem to support their factual description of fluxes, but the interpretation and expectations are unclear."

We agree with the author that OGDH acts on all of the substrate and not just glutamine-derived α -KG. As mentioned in Comment 2, since we are using labelled glutamine, if M4 fumarate, malate and citrate derived from glutamine oxidation through OGDH is reduced, it will specifically indicate reduced enzymatic activity of OGDH. Similar interpretation can be done for ACLY. To increase the clarity, we have increased the description for tracing of labeled metabolites before

describing the results. We hope that reviewer agrees that this has improved the understanding and clarity for readers.

In the revised manuscript “M5 glutamine is converted to glutamate, which is further converted into M5 α -KG by glutamate dehydrogenase or transaminases²⁶” is replaced by “M5 glutamine is converted into M5 glutamate, which is further converted into M5 α -KG by glutamate dehydrogenase or transaminases²⁶. The M5 α -KG is decarbonized into M4 fumarate, M4 malate, and M4 citrate, obtained by condensing of M4 oxaloacetate obtained from glutamine and unlabeled acetyl CoA from glucose.”

The part “We then used [U-¹³C₆] glucose to trace its contribution to intermediate metabolites within the TCA cycle (Fig. 6i).” is replaced by “We then used [U-¹³C₆] glucose to trace its contribution to intermediate metabolites within the TCA cycle (Fig. 6i). The [U-¹³C₆] glucose will be converted into M3 pyruvate through glycolysis, and M3 pyruvate will be further converted into M2 acetyl-CoA, and M2 TCA cycle metabolites (citrate, α -KG, fumarate, and malate)”

4. “It seems like the authors are ultimately talking about a shift from glutaminolysis to use of glucose with USP13 knockdown, but are trying to mechanistically explain it via the action of a TCA enzyme. This needs to be explained better/more clearly.”

As mentioned before, our new data under knock down conditions of ACLY, OGDH and double knock down of OGDH and ACLY clarifies the role of USP13 in regulating TCA cycle metabolites and lipogenesis. Further, we have now added glucose labeling results where we show that glycolysis is increased during USP13 knockdown and further glucose's contribution to TCA is reduced.

5. “And overall, the story seems to be saying that there is overexpression of USP13 in OVCA, which is causing deubiquitination and thus removal of ubiquitin that would otherwise target OGDH for degradation, and thus more OGDH, and thus more TCA cycle enzymes... this seems to be in contrast to the typical expectation that oxphos is not the dominant mode in cancer cells. This should be addressed more head-on and reconciled, perhaps in the discussion.”

We thank the reviewer for bringing up this point. Actually, recent studies in ovarian and in many cancers indicate that mitochondrial function is not downregulated in cancer cells. We recently

showed that aggressive ovarian cancer cells dominantly rely on glutamine driven mitochondrial metabolism for their invasion and metastasis. We have modified the discussion to address this concern. Please see below for the modification.

Originally we wrote that “Metabolic rewiring in cancer cells may render them highly dependent on specific metabolic enzymes or processes, which can be exploited for cancer therapy⁴⁸. However, there continue to be challenges to target cancer metabolism for treatment in understanding which metabolic pathways are altered in cancer². With the availability of multi-level databases in cancer genomics, transcriptomics, and proteomics on the horizon for clinical care, a starting point now is to identify genomic events that frequently occur and play causal roles in a particular type of cancer. In this study, we identified frequent copy number gain of the *USP13* gene in OVCA and found two USP13 deubiquitination targets that determine glutaminolysis, mitochondrial respiration, and lipid synthesis. Our findings suggest that USP13 amplification is likely an important driver in ovarian tumour progression. Amplification of USP13 allows OVCA cells to rely on glutamine anaplerosis to replenish the TCA cycle with metabolic intermediates. Moreover, increased level of USP13 promotes core metabolic pathways to generate ATP, reducing equivalents and main precursors for lipid biosynthesis. Our study provides a potential therapeutic strategy in which targeting USP13 blocks biosynthesis of metabolic intermediates and lipids thereby simultaneously inducing energy stress and cell death.”,

In the revised manuscript we have modified above to:

“Metabolic rewiring in cancer cells may render them highly dependent on specific metabolic enzymes or processes, which can be exploited for cancer therapy⁴⁸. However, there continue to be challenges to target cancer metabolism for treatment in understanding which metabolic pathways are altered in cancer². With the availability of multi-level databases in cancer genomics, transcriptomics, and proteomics on the horizon for clinical care, a starting point now is to identify genomic events that frequently occur and play causal roles in a particular type of cancer. In this study, we identified frequent copy number gain of the *USP13* gene in OVCA and found two USP13 deubiquitination targets that determine glutaminolysis, glucose oxidation, mitochondrial respiration, and lipid synthesis. Our findings suggest that USP13 amplification is likely an important driver in ovarian tumour progression. Amplification of USP13 allows OVCA cells to rely on glutamine anaplerosis to replenish the TCA cycle with metabolic intermediates, and USP13 knocked down represses mitochondrial function. The biogenesis of mitochondria,

and its integration of metabolism and cellular signalling in cancer cells has been proven to play an important role in cancer initiation, progression, and metastasis (Weinberg et al., 2010; LeBleu et al., 2014; Yang et al., 2014; Zong et al., 2016). Moreover, increased level of USP13 promotes core metabolic pathways to generate ATP, reducing equivalents and main precursors for lipid biosynthesis. Our study provides a potential therapeutic strategy in which targeting USP13 blocks biosynthesis of metabolic intermediates and lipids thereby simultaneously inducing energy stress and cell death.

References:

- Weinberg, F., Hamanaka, R., Wheaton, W.W., Weinberg, S., Joseph, J., Lopez, M., Kalyanaraman, B., Mutlu, G.M., Budinger, G.R.S., and Chandel, N.S. (2010). Mitochondrial metabolism and ROS generation are essential for Kras-mediated tumorigenicity. *Proc. Natl. Acad. Sci. U.S.A.* *107*, 8788–8793.
- Yang, L., Moss, T., Mangala, L.S., Marini, J., Zhao, H., Wahlig, S., Armaiz-Pena, G., Jiang, D., Achreja, A., Win, J., et al. (2014). Metabolic shifts toward glutamine regulate tumor growth, invasion and bioenergetics in ovarian cancer. *Molecular Systems Biology* *10*.
- LeBleu, V.S., O'Connell, J.T., Gonzalez Herrera, K.N., Wikman, H., Pantel, K., Haigis, M.C., de Carvalho, F.M., Damascena, A., Domingos Chinen, L.T., Rocha, R.M., et al. (2014). PGC-1 α mediates mitochondrial biogenesis and oxidative phosphorylation in cancer cells to promote metastasis. *Nat Cell Biol* *16*, 992–1003.
- Zong, W.-X., Rabinowitz, J.D., and White, E. (2016). Mitochondria and Cancer. *Molecular Cell* *61*, 667–676.

Reviewer#4:

1. *"...The authors propose a therapeutic strategy for HGSC involving combined inhibition of USP13 and the PI3K/AKT pathway. The study was comprehensive and findings were clearly and logically presented."*

We thank the reviewer for his/her positive comments. In the revised manuscript, we have performed a number of additional experiments to provide further evidence to support our hypothesis and address all the review comments.

2. *"USP13 protein levels were analyzed by IHC on TMA. In figure 1e, were the adjacent normal samples and HGSC samples presented matched? Y-axis is labeled as intensity but the values appear to be a combination of intensity and % cells stained. Authors should define 'a.u.' (arbitrary units?)"*

In Fig. 1e, we presented HGSC samples and their matched normal tissues. The values on Y-axis is the combination of intensity and % cells stained, therefore we have corrected the Y-axis label to USP13 level (a.u.) as suggested and defined 'a.u.' as arbitrary units in the figure legend. In addition, we have also performed Western blot results from ovarian tumor and normal tissue samples in Fig. 1f.

3. *“Authors should indicate if control constructs used in shRNA experiments were empty vector or scramble?”*

Thank you for pointing out this issue. Control constructs in all of the studies express scramble shRNAs. We have pointed out in the Figure Legends and Methods.

4. *“In Figure 2C, are the clonogenic assay images shown from KD-1 or KD-2? At what time point where colonies counted? Methods should be ordered in line with Results.”*

We apologize for our sloppiness. All the representative clonogenic assay images shown in Fig. 2c are from KD-1. We counted colony numbers at day 7. Methods have been rewritten and ordered in line with results in the revised manuscript.

5. *“Has PIK3CA and/or pathway member protein expression been characterized in cell lines? Is PIK3CA amplification or PI3K/AKT pathway activation associated with MK2206 sensitivity? It is not clear why the authors chose to look at a pan-AKT inhibitor rather than a PI3K inhibitor in the first instance.”*

In our study, we found that *PIK3CA* gene copy number is not tightly correlated with its protein level (Supplementary Fig. 3e and TCGA database analysis not shown). However, consistent with a previous study (Cancer Discov. 2012, 2(1): 56-67), we also found that PI3K/AKT pathway activation (indicated by pAKT levels in Supplementary Fig. S3e) is closely associated with the sensitivity of ovarian cancer cells to the treatment of MK2206. Therefore, we chose to use the pan-AKT inhibitor MK2206 in our studies to test if inhibiting USP13 enhances the sensitivity of ovarian cancer cells.

6. *“It should be noted that HeyA8, A2780, SKOV3 are not considered to be representative of HGSC (Domcke et al., 2015).”*

We fully agree with the reviewer and have also noticed this recent paper clarifying the tumour origin of popular ovarian cancer cell lines. To address this concern, we have also used two HGSC cell lines OVCAR3 and OAW28 in our studies and observed the same effect of USP13 inhibition on cell viability. The results are included in Fig. 2b and Supplementary Fig. 3d.

7. *“Error bars should be presented in Figure 7 c and N should be reported for 7d.”*

As suggested, we have added a table in Supplementary Fig. 8b to present mouse numbers and *p* values. Significant difference in the metastatic patterns of the control and USP13 knockdown groups were compared by Fisher’s exact test. In Fig. 7c, the percentage of mice with metastasis in a total of 10 tested mice was counted, therefore, no error bars could be presented.

8. *“Were markers other than Ki67 quantitated for data presented in Figure 7h”*

In addition to Ki67, we have also measured the levels of AKT and pAKT levels in tumour samples in the new Fig. 9d.

9. *“Can the authors conclude that the effect of USP13 suppression by shRNA KD and MK2206 treatment is synergistic rather than additive? Has synergy between MK2206 and USP13 inhibitors been assessed (e.g. using Chou-Talalay methodology)?”*

It is a great point from the reviewer. We have used Choi-Talalay methodology to assess the synergy between MK2206 and USP13 inhibitors in cytotoxicity of ovarian cancer cells (Fig. 2e). To use Choi-Talalay method, multiple points of USP13 expression level from low to high need to be achieved by Dox-inducible inhibition, which was technically challenging. We could not have USP13 expressed at gradient levels in our experiment. Therefore, it is now premature to assess the synergy between MK2206 and USP13 inhibition. My laboratory now is going to screen and identify small molecule inhibitor against USP13, which will be used in the future for this type of studies.

Thank you again for all of your thoughtful and helpful comments. By addressing these concerns, we have considerably strengthened the manuscript and hope you agree.

REVIEWERS' COMMENTS:

Reviewer #1 (Remarks to the Author):

Han et al. explored the role of de-ubiquitinase USP13 in ovarian cancer metabolic regulation and tumor progression. They found that the USP13 gene is frequently amplified in human OVCA, and its overexpression is significantly associated with poor clinical outcome. Consistently, they demonstrated that USP13 silencing inhibited ovarian cancer growth in vitro and in vivo by specifically deubiquitinating two critical enzymes named ATP citrate lyase (ACLY) and oxoglutarate dehydrogenase (OGDH), leading to mitochondrial respiration, glutaminolysis, and fatty acid synthesis. Notably, inhibiting USP13 sensitized OVCA cells to the treatment of AKT inhibitor, suggesting that combined treatment of PI3K/AKT inhibitor with USP13 inhibitor might be a promising targeting strategy to overcome ovarian cancer. This work unravels a novel link between USP13 and ovarian cancer by modulating tumor metabolism. The authors have adequately addressed all of my concerns, and the study is now thorough, extensive and appropriate for its publication at NCB.

Reviewer #2 (Remarks to the Author):

The authors have sufficiently addressed my critics. The manuscript was further strengthened and appropriate to publish.

Reviewer #3 (Remarks to the Author):

After re-reading the authors' responses a number of times and directly comparing this draft to the previous one, I believe the authors have addressed all of my concerns from the previous review. However, the way in which they did so could have been much more straightforward and transparent. Their conclusions have changed in some noteworthy respects based on the additional experiments that they did, and they did not make this clear. (They made it clear that they included additional results, and in some cases indicated some of the conclusions drawn from the additional results, but they did NOT make it explicit that some of the intermediate interpretations were changing.) It was only upon going back to the original version and comparing it paragraph-by-paragraph to the current version that I was able to identify the key changes. The results of those changes seem reasonable --- the changes in conclusions about what metabolic changes happen do not affect the overall story, so that is not a concern. The additional experiments seem to support their new conclusions. But the authors could have been more upfront about this, particularly because in responses to some of my previous comments, they explicitly included "old" and "new" text to show the changes they made. While that degree of detail isn't necessary all of the time, clearly stating when conclusions change should always be done; otherwise one might be concerned that they are intentionally obfuscating relative to their other explanations. The additional explanations in the text regarding interpretation of different M2/3/4/5 analytes was definitely helpful and appreciated.

Reviewer #4 (Remarks to the Author):

The authors have satisfactorily addressed comments and concerns raised.

Response to Reviewers (Revised manuscript NCOMMS-16-03474-A)

We are very glad to know that our manuscript entitled “Amplification of USP13 drives ovarian cancer metabolism” is accepted in principle. We thank you for your helpful comments in the last review and we are happy to know that all the review comments have been addressed.

Sincerely,

Xiongbn Lu, Ph.D.
Department of Cancer Biology
University of Texas MD Anderson Cancer Center